# The Structural Role of RPN10 in the 26S Proteasome and an RPN2-Binding Residue on RPN13 Are Functionally Important in Arabidopsis

**DOI:** 10.3390/ijms252111650

**Published:** 2024-10-30

**Authors:** Shih-Yun Lin, Ya-Ling Lin, Raju Usharani, Ramalingam Radjacommare, Hongyong Fu

**Affiliations:** 1Institute of Plant and Microbial Biology, Academia Sinica, Taipei 115, Taiwan; linsiun@gmail.com (S.-Y.L.); usharaj1983@gmail.com (R.U.); rradjacommare@rediffmail.com (R.R.); 2Program in Biological and Sustainable Technology, Academy of Circular Economy, National Chung Hsing University, Nantou 540, Taiwan; yalinglin@dragon.nchu.edu.tw

**Keywords:** RPN10, RPN13, RPN2, UCH1, UCH2, ECM29, PA200, ubiquitin receptor, 26S proteasome, Arabidopsis

## Abstract

The ubiquitin receptors RPN10 and RPN13 harbor multiple activities including ubiquitin binding; however, solid evidence connecting a particular activity to specific in vivo functions is scarce. Through complementation, the ubiquitin-binding site-truncated Arabidopsis RPN10 (N215) rescued the growth defects of *rpn10-2*, supporting the idea that the ubiquitin-binding ability of RPN10 is dispensable and N215, which harbors a vWA domain, is fully functional. Instead, a structural role played by RPN10 in the 26S proteasomes is likely vital in vivo. A site-specific variant, RPN10-11A, that likely has a destabilized vWA domain could partially rescue the *rpn10-2* growth defects and is not integrated into 26S proteasomes. Native polyacrylamide gel electrophoresis and mass spectrometry with *rpn10-2* 26S proteasomes showed that the loss of RPN10 reduced the abundance of double-capped proteasomes, induced the integration of specific subunit paralogues, and increased the association of ECM29, a well-known factor critical for quality checkpoints by binding and inhibiting aberrant proteasomes. Extensive Y2H and GST-pulldown analyses identified RPN2-binding residues on RPN13 that overlapped with ubiquitin-binding and UCH2-binding sites in the RPN13 C-terminus (246–254). Interestingly, an analysis of homozygous *rpn10-2* segregation in a *rpn13-1* background harboring RPN13 variants defective for ubiquitin binding and/or RPN2 binding supports the criticality of the RPN13–RPN2 association in vivo.

## 1. Introduction

Ubiquitination, the post-translational modification of cellular proteins via the covalent conjugation of ubiquitin monomers or polymers of various lengths and linkages, creates a set of crucial signal codes for numerous cellular regulatory mechanisms. These include, for example, proteasome-mediated proteolysis, endocytosis, signal transduction, transcriptional activation/suppression, mRNA splicing/export, autophagy, and chromatin activation/silencing [1]. Numerous essential regulatory proteins that are involved in different aspects of plant growth, development, biotic and abiotic stress responses, signaling pathways of various plant hormones, self-incompatibility, and photomorphogenesis are meticulously tuned by ubiquitination and targeted for subsequent ubiquitin-dependent regulatory processes, including 26S proteasome-mediated degradation [2]. 

Ubiquitinated proteins are targeted to 26S proteasomes via a set of evolutionarily conserved ubiquitin receptors, including intrinsic proteasome subunits RPN1, RPN10, and RPN13, as well as extrinsic ubiquitin-like/ubiquitin-associated (UBL-UBA) shuttle factors [3,4]. Interestingly, extrinsic UBL-UBA shuttle factors are often recruited to 26S proteasomes by intrinsic ubiquitin receptors [3,4,5,6]. The involvement and interconnectivity of multiple intrinsic and extrinsic ubiquitin receptors supports a highly coordinated and cooperative recognition and processing mechanism for ubiquitinated substrates by 26S proteasomes [3,4,6,7,8,9]. However, the multiplicity and redundancy of ubiquitin receptors have made deciphering the in vivo functions an arduous task for each recognition pathway [3,10,11].

RPN10 was the first ubiquitin recognition factor to be discovered [12,13]. It is an intrinsic subunit of the 26S proteasome that harbors an N-terminal von Willebrand factor type A (*vWA*)-like domain and 1–3 C-terminal Ubiquitin-Interacting Motifs (UIMs), which are critical for lid–base association [14,15] and the direct/indirect recognition of ubiquitinated substrates [5,10], respectively. Although deletion mutants of *RPN10* display mild to severe growth and developmental defects in yeast [16], mice [17], Drosophila [18], and Arabidopsis [5,10,19], direct evidence associating these defects with loss of the recognition activity of RPN10 for ubiquitinated substrates is scarce. Moreover, the mild flaws in the turnover of aberrant and ubiquitin fusion degradation (UFD) substrates associated with the yeast Δ*rpn10* mutant were completely rescued by a C-terminally truncated or site-specific mutated Rpn10 variant that has a defect in ubiquitin binding [20]. In contrast, several site-specific mutations in the N-terminal region surrounding the D11 residue, which is critical for the structural stability of the vWA domain and stable lid–base association, and a minimal N-terminal deletion variant (Δ1–15) of yeast Rpn10 compromised Rpn10’s ability to rescue the mild Δ*rpn10* defects [14]. 

For Arabidopsis RPN10, moderate or extremely severe vegetative and reproductive growth defects have been observed with the T-DNA insertion mutants *rpn10-1* [21] and *rpn10-2* [5,10], respectively. Surprisingly, all severe *rpn10-2* growth defects could be fully rescued by the RPN10 variant u123, which carries site-specific mutations for all three UIMs, compromising both the direct and indirect recognition of ubiquitinated substrates [10]. In contrast, a potential structural defect of 26S proteasomes was demonstrated by a significant reduction in double-capped 20S proteasomes in *rpn10-2* plants [10]. Together, the accumulated evidence indicates that the observed growth defects associated with RPN10 null mutants in various species are likely not due to compromised substrate recognition. This means that the specific in vivo roles of substrate recognition for RPN10, and for various intrinsic and extrinsic ubiquitin receptors, remain unresolved. Moreover, the potential structural role of the N-terminal domain of Arabidopsis RPN10 in 26S proteasome holoenzymes in vivo warrants further investigation. 

Established as a novel intrinsic ubiquitin receptor [6,11], RPN13 is a relatively recently discovered integral subunit [22] of the 26S proteasome. Although the RPN13 orthologue, Xoom, was found to be required for embryonic development in frogs [23], no discernible—or only mild—defects were associated with null mutants in yeast, humans, and Arabidopsis [6,10,19,24]. Initially, RPN13 was found to play a major role in the recruitment and activation of UCH37, a dominant 26S proteasome-associated deubiquitination enzyme [25,26,27]. To recruit UCH37, RPN13 first associates with the core base subunit RPN2 for assembly into the 26S proteasome [6,11,27,28]. Apart from a large C-terminal truncation with a budding yeast orthologue, RPN13 orthologues from different species, including Arabidopsis [5,19], are highly conserved, harboring an N-terminal PRU (Pleckstrin-like Receptor for Ubiquitin) and C-terminal DEUBAD (DEUBiquitinase adaptor) domains. These are critical for the perception of ubiquitinated substrates [6,11] and for the recruitment/activation of UCH37 [25,26,27], respectively. The DEUBAD domain, present in RPN13 orthologues from higher eukaryotes but not in the C-terminally shorter yeast orthologue, appears to have evolved to play a major role in the proteasomal recruitment and activation of UCH37 [27].

In addition to recruiting ubiquitinated substrates, the N-terminal PRU domain of RPN13 is also critical for RPN2 binding [25,26,27]. The structure of the PRU domain of murine RPN13, which is nearly identical to that of humans (hRPN13), has been determined by X-ray crystallography; its ubiquitin recognition interface has been mapped by NMR and molecular docking [11]. The PRU domain of RPN13 contains a novel ubiquitin-binding fold, a seven-stranded β-sandwich structure capped by a C-terminal α-helix [6,11]. The ubiquitin-binding interface located on loops across a large portion of the RPN13 PRU domain (rather than the simple secondary structural elements generally employed by UIM- and UBA-containing ubiquitin receptors) is used to capture ubiquitin signals. Some of the critical residues (e.g., S55, L56, F76, I75, D79, and F98) on the ubiquitin-binding interface of mRPN13 have been confirmed via pull-down analyses with site-specific mutants [11]. When examined via NMR titration, the RPN2-interacting interface on the PRU domain of hRPN13 appeared to be independent from the domain involved in ubiquitin recognition. Some critical residues, such as M31, C88, and E111, are located in a sector opposite the ubiquitin-binding interface [11,28]. The full-length hRpn13 structure determined by NMR shows that the C-terminal UCH37-binding DEUBAD domain forms a helical bundle packed against the ubiquitin-binding PRU domain. It appears that docking to the 26S proteasome interrupts interdomain interactions, activating both ubiquitin-binding and UCH37 recruitment activities [28]. 

The recruitment of UCH37 by RPN13 is primarily mediated by binding the C-terminal UCH37-like domain (ULD) of UCH37 to the DEUBAD domain of RPN13 [29]. The UCH37-binding interface on the hRPN13 DEUBAD domain has been mapped via NMR and further tested by pull-down assays using hRPN13 variants with alanine substitutions in two regions (V286/D287/E295/I296 and K371/D373/E375/K379) [28], which are rich in charged residues and complement the previously determined RPN13-binding region in UCH37 (Val313-Lys329) [25]. Additional hydrophobic contacts between the RPN13 C-terminal domain (285–386) and UCH37 C-terminal helices have also been described [29,30]. Structural analyses revealed that the activation of UCH37 by RPN13 binding is facilitated by precisely positioning its C-terminal UCH37-like domain (ULD) domain and crossover loop (CL) in the UCH domain to promote ubiquitin–substrate binding and catalysis [29,30]. ULD and UCH catalytic domain (CD) anchoring is achieved via polar interactions centered on a highly conserved glutamate residue, Glu284 [30], and the positioning of CL is mediated by binding to RPN13-DEUBAD [30].

Two potential Arabidopsis orthologues for human *UCH37* have been characterized: *UCH1* and *UCH2*. A *UCH1* over-expresser and a *uch1 uch2* double mutant, but not *uch1* and *uch2* single mutants, displayed an altered shoot architecture and sensitivity to auxin/cytokinin. These results suggest that UCH1 and UCH2 play redundant roles in auxin/cytokinin signaling, potentially by targeting upstream regulatory proteins such as AUXs/IAAs [31]. In addition, Arabidopsis UCH1 and UCH2, together with UCH3, which is more distantly related to human UCH37 and truncated at the C-terminal ULD, show redundant roles in clock control at elevated temperatures [32]. However, unlike their counterparts in other eukaryotes [33,34], Arabidopsis UCH1 and UCH2 were not found to be associated with 26S proteasomes prepared under mild conditions [31]. Thus, UCH1 and/or UCH2 recruitment by 26S proteasomes and their in vivo functions as an integral component may not be conserved in Arabidopsis. Further investigation is required to confirm this. 

Although the critical structural elements of RPN10 and RPN13 for ubiquitin recognition, RPN2 association, or UCH37 recruitment have been fully characterized, the in vivo importance of these elements has not been confirmed. While no clearly discernible defects were observed with the Arabidopsis RPN13 null mutant [10,19], we observed a 2% homozygous *rpn10-2* segregation rate loss with heterozygous *rpn10-2* plants when segregation was conducted in a homozygous *rpn13-1* background [19]. In this study, using *rpn10-2*, we examined the in vivo importance of the RPN10 N-terminal domain [10]. We determined the same for RPN13 structural elements by examining the segregation of homozygous *rpn10-2* from heterozygous *rpn10-2* in a homozygous *rpn13-1* background harboring RPN13 variants with mutations of critical elements for ubiquitin recognition and RPN2 association.

## 2. Results

### 2.1. N-Terminal Portion (1–215) of Arabidopsis RPN10 (N215) Is Fully Functional In Vivo

Homozygous Arabidopsis *rpn10-2* mutant plants, segregated from heterozygous *rpn10-2* plants at a rate of approximately 2%, were sterile and displayed various severe vegetative and reproductive growth defects [10]. The ubiquitin receptor activities of RPN10 appear to be nonessential for in vivo functions, as all the examined growth defects of the null mutant *rpn10-2* could be completely rescued by complementation with an RPN10 variant, RPN10-u123 (u123). This variant is mutated at multiple sites in each of the three UIMs responsible for the direct and indirect recognition of ubiquitinated substrates [5,10]. To further delineate the domains and residues essential for the in vivo functions of RPN10, we performed complementation experiments with heterozygous *rpn10-2* plants using each of the following constructs encoding deletion or site-specific mutated RPN10 variants: three N-terminal deletions (*NΔ15*, *NΔ42*, and *NΔ53*); one C-terminal deletion (*N215*); and several N-terminal site-specific mutated variants (*11A*, *11R*, *11K*, *11*–*12RR*, and *10*–*14A5*) (Figure 1A). The *rpn10-2* heterozygous plants were transformed independently with each of the above-described *RPN10* constructs. T3 plants homozygous for the *rpn10-2* allele and any of the *RPN10* variants were segregated and could only be generated if the reproductive defects of *rpn10-2* were rescued by a specific functional RPN10 variant. 

To confirm that the C-terminal region of RPN10 harboring the UIM1, UIM2, and UIM3 motifs was not essential for RPN10’s various in vivo functions [10], we first performed a complementation experiment with heterozygous *rpn10-2* using the *N215* construct, which only encodes the N-terminal region (1–215) harboring a putative vWA domain [35] but not the C-terminal region (216–386) harboring the UIM1, UIM2, and UIM3 motifs (Figure 1A). Several independent T3 lines (*N215-11*, *N215-12*, *N215-14*, and *N215-16*), homozygous for both the *rpn10-2* and *N215* alleles, were easily generated, indicating that the N-terminal RPN10 fragment (1–215) rescued the reproductive growth defects of the *rpn10-2* null mutant. Surprisingly, N215 protein expression was not detected in any of the examined *N215* lines (Figure 1B). This indicates that extremely low expression levels of N215, which were not detected with our polyclonal antisera against Arabidopsis RPN10, can adequately rescue the reproductive growth defects associated with *rpn10-2*. Alternatively, N215 could be missing the epitopes that allow it to be detected by the α-RPN10 antisera. 

We examined whether *N215* could rescue the major vegetative and, in particular, reproductive growth defects of homozygous *rpn10-2* in *N215*-complemented *rpn10-2* lines. We first confirmed the absence of RPN10 in the null mutant *rpn10-2* (Figure 1B). In comparison with Col-0, the significant induction of prominent 20S proteasome subunits (the identities of which were not determined) were also observed, as previously described [10]. In agreement with the previously described results [10,19], the homozygous *rpn10-2* showed various growth phenotypes in comparison with Col-0 (Figure 1 and Appendix A, *rpn10-2* vs. Col-0). The confirmed phenotypes of *rpn10-2* included delayed dark-induced senescence (Figure 1C); an altered percentage distribution of rosette leaf trichomes with two to five branches (5, 58, 31, and 6% vs. 8, 86, 6, and 0%, respectively, for rosette leaf trichomes of *rpn10-2* and Col-0 with two, three, four, and five branches) (Figure 1D); short seedling primary root lengths (Appendix A); late flowering, measured either as DAS (days after stratification) or rosette leaf number at bolting (Figure 1E); increased pedicel lengths (Figure 1F and Appendix A); large flowers with larger petal areas (Figure 1G and Appendix A); and abnormal floral organ numbers (Appendix A). Homozygous *rpn10-2* plants bearing under-developed short siliques with complete (100%) ovule abortion were clearly sterile (Figure 1H and Appendix A). They showed an increased final plant height, likely due to extended vegetative growth (Appendix A). The *rpn10-2* short siliques, however, displayed slightly increased ovule numbers per carpel (Appendix A). 

N215 appeared to be capable of rescuing all the homozygous *rpn10-2* growth defects, as described above (Figure 1 and Appendix A, *N215* lines). Consistent with previous findings, the induction of 20S proteasomes was alleviated and the restoration extent (but not to the wild-type level) was generally proportional to the expression levels or rescue capabilities of the introduced RPN10 variant [10]; an exception was *u123*-complemented *rpn10-2* lines which showed high levels of 20S proteasomes. Interestingly, in agreement with these findings, the induced prominent 20S proteasome subunits in all N215 lines were restored to similar levels as that of cN10-20 (a previously described wild-type RPN10-complemented *rpn10-2* line [10]). 

Similar to *cN10*- and *u123*-complemented *rpn10-2* lines (*cN10-20* and *u123-3*) described previously [10], two to four *N215*-complemented *rpn10-2* lines were shown to rescue all the examined *rpn10-2* growth defects. They displayed various phenotypes generally similar to those of Col-0 plants: normal dark-induced leaf senescence (Figure 1C, *N215-11* and *N215-12*); and similar percentage distributions of leaf trichomes with two–five branches (Figure 1D); seedling primary root lengths (Appendix A); final plant heights (Appendix A); flowering times measured either as DAS or rosette leaf number at bolting (Figure 1E); pedicel lengths (Figure 1F and Appendix A); silique lengths (Appendix A); flower and petal sizes (Figure 1G and Appendix A); floral organ numbers (Appendix A); ovule numbers per carpel (Appendix A); and ovule abortion rates (Figure 1H). These results support the assertion that the N-terminal portion (1–215) of Arabidopsis RPN10 (N215), which harbors a vWA domain but is missing all ubiquitin-binding motifs, is fully functional in vivo. They also confirm the previous conclusion that the direct and indirect ubiquitin receptor activities of RPN10 are not essential for the in vivo functions observed in the homozygous *rpn10-2* mutant [10]. 

### 2.2. The D11 Residue of RPN10 Plays a Critical Role In Vivo and for Assembly into the 26S Proteasome

To identify the functionally essential domains and residues located within the RPN10 N-terminal portion, complementation experiments were performed using heterozygous *rpn10-2* plants and constructs encoding each of the above-described N-terminal deletion and site-specific mutated RPN10 variants: *NΔ15*, *NΔ42*, *NΔ53*, *11A*, *11R*, *11K*, *11*–*12RR*, and *10*–*14A5* (Figure 1A). However, through repeated efforts, T3 lines double homozygous for the *rpn10-2* allele and a specific *RPN10* N-terminal variant could only be segregated with *11A* and not with any other construct. RPN10-11A variant expression levels from five established *11A*-complemented *rpn10-2* lines (Figure 1B, *11A-1*, *11A-2*, *11A-3*, *11A-10*, and *11A-12*) were slightly higher than those of u123 from *u123-3* and nearly equivalent to those of RPN10 in *cN10-20* (as described previously [10] and shown above (Figure 1 and Appendix A)), which completely rescued the examined *rpn10-2* defects. 

As T3 plants homozygous for both *rpn10-2* and *RPN10-11*A alleles could be segregated, the reproductive defects of the *rpn10-2* mutant might have been rescued by RPN10-11A. However, in contrast to the *cN10-20*, *u123-3*, and *N215* lines, which were shown to completely rescue of all the *rpn10-2* growth defects examined, the *RPN10-11A*-complemented *rpn10-2* lines were shown to have only partially rescued these growth defects including fertility. Interestingly, an intermediate induction of 20S proteasome subunits in all *A11* lines was observed. The induction levels were restored to levels slightly lower than those of *rpn10-2* but higher than those of the cN10-20 and N215 lines (Figure 1B). The plants of two to five *RPN10-11A* lines generally displayed intermediate phenotypes between the severe growth defects observed in homozygous *rpn10-2* plants and the wild-type growth characteristics of Col-0, as well as those of fully rescued *cN10*-, *u123*-, and *N215*-complemented *rpn10-2* lines (Figure 1 and Appendix A, *11A* lines). In contrast to the rosette leaves of Col-0, as well as *cN10*- and *N215*-complemented *rpn10-2* lines, which showed normal dark-induced senescence, the rosette leaves of two *RPN10-11A*-complemented *rpn10-2* lines showed slightly delayed dark-induced senescence to different extents (Figure 1C, *11A-1* and *11A-2*). Decreased or increased percentages of leaf trichomes with different branch numbers for *11A*-complemented *rpn10-2* appeared to be intermediate between those of *rpn10-2* and Col-0, as well as those of fully rescued *cN10*-, *u123*-, and *N215*-complemented *rpn10-2* lines (Figure 1D). As examples, the average percentages for leaf trichomes of the *11A* lines with three and four branches were 79 and 14%, respectively, which are intermediate between the numbers of *rpn10-2* (58 and 31%) and Col-0, as well as *cN10*-, *u123*-, and *N215*-complemented *rpn10-2* lines (ranging between 82 and 87 and between 5 and 10%). Moreover, the *RPN10-11A*-complemented *rpn10-2* lines also displayed intermediate growth phenotypes between those of *rpn10-2* and Col-0, as well as those of fully rescued *rpn10-2* lines (complemented with *cN10*, *u123*, and *N215* lines) for seedling primary root lengths (Appendix A), final plant heights (Appendix A), flowering times when measured as DAS at bolting (Figure 1E), pedicel lengths (Figure 1F and Appendix A), silique growth (Appendix A), flower and petal sizes (Figure 1G and Appendix A), and average ovule abortion rates (Figure 1H). Similarly, while the abnormal petal numbers associated with *rpn10-2* were generally restored (11A-complemented lines), the observed abnormal stamen numbers remained similar to those of *rpn10-2* plants (Appendix A). However, similar to all rescued *rpn10-2* lines (*cN10*-, *u123*-, and *N215*-lines), all *11A* lines generally showed average ovule numbers similar to that of wild-type plants (Appendix A).

In agreement with a previous report [10], when examined by native PAGE followed by an in-gel activity assay and immunoblotting, the abundance of double-capped 20S core particles was drastically reduced in partially purified 26S proteasomes from *rpn10-2* in comparison with those from Col-0 (Figure 2, top two panels). Although less than that from Col-0, the abundance of double-capped 20S core particles was clearly increased for 26S proteasomes prepared from *cN10*-, *u123*-, and *N215*-complemented *rpn10-2* lines in comparison with that of *rpn10-2* (Figure 2, top two panels). Correlating with the partially compromised growth phenotypes, the abundance of double-capped 20S core particles in 26S proteasomes purified from *RPN10-11A*-complemented *rpn10-2* lines was intermediate between that of *rpn10-2* and Col-0, as well as *cN10*-, *u123*-, and *N215*-complemented *rpn10-2* lines. However, the abundances of single-capped 20S core particles were similar among the partially purified 26S proteasomes from all the examined lines. 

We examined whether the assembly of any RPN10 variant into single- and double-capped 20S core particles was affected using partially purified 26S proteasomes from the examined lines. Using native PAGE, both wild-type RPN10 and u123 were clearly detected by immunoblotting in both single- and double-capped 20S core particles of the 26S proteasomes partially purified from Col-0, *cN10*-, or *u123*-complemented *rpn10-2* lines but, as expected, were not detected in both complexes of the 26S proteasomes from the null *rpn10-2* line (Figure 2, α-RPN10). Particularly interesting is that RPN10-11A was not detected in both the single- and double-capped 20S core particles of the 26S proteasomes prepared from the *11A*-complemented *rpn10-2* lines, indicating that the N-terminal D11 residue of RPN10 is critical for its assembly in this regard. In agreement with the undetectable expression (or missing detectable epitopes for α-RPN10) of N215 in *N215*-complemented *rpn10-2* lines, N215 was also not detected in both single- and double-capped 20S core particles (Figure 2, α-RPN10).

### 2.3. Presence of Specific Paralogues for Several Core Subunits and Increased ECM29 and PA200 Protein Levels in Single-Capped 20S Proteasomes from rpn10-2

The critical role of D11 in RPN10’s integration into the 26S proteasome, as well as the correlation between the reduced abundance of double-capped 20S proteasomes and growth defects in *rpn10-2* and various RPN10 variant-complemented *rpn10-2* lines suggest that there are potential conformational changes or structural defects in the 26S proteasomes, both single- and double-capped 20S proteasomes, in *rpn10-2*. As a first step in exploring this plausibility, we collected single-capped 20S proteasomes after native gel electrophoresis of 26S proteasomes partially purified from Col-0 and *rpn10-2* and subjected them to mass spectrometry analyses. As shown in Table 1, the identified subunits generally agree with previous mass spectrometry analyses of Arabidopsis 26S proteasomes prepared using polyethylene glycol (PEG) precipitation together with anion exchange and size exclusion chromatography [36] or affinity purification methods [37,38]. Except for one core particle (CP) subunit (PAC2), four base subunits (RPT1b, RPT2a, RPN13, and RPN15), and one lid subunit (RPN12b), the rest of the 26S proteasome subunits could be detected in the single-capped 20S proteasomes of Col-0 or *rpn10-2* (Table 1).

Based on the quantification of the normalized total number of PSMs (peptide spectrum matches), five CP subunits (PAB2, PAF2, PBB2, PBC2, and PBE2), two base subunits (RPT5b and RPT6b), and one lid subunit (RPN8b) were not detected in the single-capped 20S proteasomes of Col-0. In general, low (5.2–17.3) and median (73.1 and 39.5 for RPT6b and RPN8b, respectively) numbers of PSMs were found for these subunits in the single-capped 20S proteasomes of *rpn10-2*. As expected, RPN10 was only detected in the single-capped 20S proteasomes of Col-0 but not in those of *rpn10-2*. Interestingly, a significant median number of collected PSMs was found for RPN9a in the single-capped 20S proteasomes of Col-0; it was not detected in those of *rpn10-2*. Moreover, in comparison with those of Col-0, a significantly increased number of collected PSMs was associated with RPN9b in the single-capped 20S proteasomes of *rpn10-2*. The mass spectrometry results suggest that a portion of the 26S proteasomes of *rpn10-2*, in comparison with those from Col-0, could potentially be composed of distinct paralogues for five CP subunits (PAB, PAF, PBB, PBC, and PBE), two base subunits (RPT5 and RPT6), and two lid subunits (RPN8 and RPN9). 

In agreement with Yang et al. [36], a subtilisin-like serine aminopeptidase TPP II was also identified as a potential 26S proteasome-associated contaminant protein with single-capped 20S proteasomes from Col-0 and *rpn10-2* (Table 1). As expected, weakly bound 26S proteasomes-associated proteins, such as ubiquitin receptors (RAD23a-d, DSK2a-b, DDI1, NUB1, CDC48a–c, p47-1–3, UFD1a–d, and NPL4a–b), the CP effector PTRE1, ubiquitin ligases (UFD2 and UPL7), and DUBs (UCH1–3, UBP6–7, and UBP16), were not detected. Similarly, almost all CP (PBAC1–5 and UMP1a–c) and RP assembly chaperones (NAS2, NAS6, and HSM3) were not detected. However, as described previously by Gemperline et al. [38], some CP (PBAC1–4 and UMP1a–b) and RP assembly chaperones (NAS2, NAS6, and HSM3) were co-purified with the 26S proteasomes prepared via a more mild affinity purification method. 

In agreement with Book et al. and Gemperline et al. [37,38], two well-known 26S proteasome-associated regulatory proteins, PA200 (Blm10 in yeast) and ECM29, involved in quality control, assembly, and/or disassembly, were also consistently detected in the single-capped 20S proteasomes from Col-0 and *rpn10-2* (Table 1). Intriguingly, the abundances of PA200 and, in particular, ECM29, were found to be significantly increased (1.7- and 2.9-fold, respectively) in the single-capped 20S proteasomes from *rpn10-2* in comparison with those of Col-0 (Table 1). 

### 2.4. Expression of Genes Encoding Subunits and Biosynthesis Components of 26S Proteasomes Was Generally Significantly Increased in rpn10-2

In yeast, plants, and metazoans, the coordinated activation, by one or multiple master transcriptional regulators for genes encoding subunits and biosynthesis components, of 26S proteasomes under proteasome stresses and in various proteasome mutants is well established (reviewed in [39]). The altered abundance of a particular paralogue for each 26S proteasome subunit in *rpn10-2* could potentially be due to distinct induction levels of the corresponding transcript caused by proteasome defects. Similarly, the increased abundances of ECM29 and PA200 in *rpn10-2* could also be caused by this enhanced expression. We measured the expression levels, in reads per kilobase per million mapped reads (RPKM), of genes encoding 26S proteasome subunits, 26S proteasome-associated proteins, UPS components, and 26S proteasome assembly chaperones from RNA-seq-based transcriptomes established from 21 DAS Col-0 and *rpn10-2* rosette leaves (the complete transcriptome analyses will be reported elsewhere). 

As shown in Appendix A, except for a few low expressors or potential pseudogenes (*PAC2*, *RPT6a*, *RPN13*, *RPN15*, *RPN9a*, and *RPN12b*), almost all of the unique and paralogous genes encoding each of the 26S proteasome subunits were significantly induced in *rpn10-2* (1.6–9.9-fold). As expected with the presence of a T-DNA insertion, the RPKM detected for *RPN10* in *rpn10-2* was only 38% in comparison with that of Col-0; clear T-DNA insertion-interrupted reads were also detected. In line with the moderate detection of RPN8b and, in particular, RPT6b proteins with the single-capped 20S proteasomes of *rpn10-2*, but not Col-0 (Table 1), drastic RPKM increases were observed for *RPN8b* and *RPT6b* in *rpn10-2* (7.1- and 9.9-fold, respectively; Appendix A). Similarly, in agreement with the fact that PAB2, PAF2, PBB2, PBC2, PBE2, and RPT5b were not detected with the single-capped 20S proteasomes of Col-0, but were detected in low abundance with those of *rpn10-2* (Table 1), low expression levels of their corresponding transcripts were observed in Col-0. In contrast, clear increases in these transcripts were seen in *rpn10-2* (Appendix A). Moreover, in agreement with the drastically increased detection of ECM29 and PA200 with the single-capped 20S proteasomes of *rpn10-2* in comparison with Col-0 and although the expressions of their encoding genes (At2g26780 and At3g13330) were at relatively low levels in Col-0 and *rpn10-2*, significant RPKM increases were observed in *rpn10-2* (2.8- and 3.2-fold, respectively; Appendix A).

As general controls, expression induction in *rpn10-2* was examined for those genes encoding weakly bound 26S proteasomes-associated proteins or transiently associated 26S proteasome-assembly chaperones such as ubiquitin receptors (RAD23a–d, DSK2a–b, DDI1, NUB1, CDC48a–c, p47-1–3, UFD1a–d, and NPL4a–b), ubiquitin ligases (UFD2 and UPL7), DUBs (UCH1–3, UBP6–7, and UBP16), the CP effector PTRE1, and assembly chaperones for CPs (PBAC1–5 and UMP1a–c) and RPs (NAS2, NAS6, and HSM3). As shown in Appendix A, among the 49 examined genes, most were expressed in both Col-0 and *rpn10-2* at relatively low levels (the exceptions being *UBQ1*, *CDC48a*, and *HSP90-7*); however, similar to the genes encoding ECM29 and PA200, significantly increased RPKM values were detected for *DSK2a*, *CDC48a–b*, *UFD1d*, *PTRE1*, *UCH3*, *UBP6–7*, *PBAC1–2*, *UMP1a*, *NAS2*, *NAS6*, and *HSM3*, suggesting that the latter genes, including those encoding endoplasmic-reticulum-associated degradation (ERAD) components and 26S proteasome-assembly chaperones, also undergo coordinated transcriptional induction under proteasomal stresses and in mutants. The coordinated induction under proteasomal stresses and in mutants for genes encoding ECM29 and various 26S proteasome-assembly chaperones, including PBAC1–2, UMP1a, NAS2, NAS6, and HSM3, have also been observed previously [38,40]. Interestingly, *Saccharomyces cerevisiae* ECM29 also undergoes an RPN4-mediated transcriptional feedback regulation [41]. 

### 2.5. RPN13 Is Associated with RPN2 and UCH2 but Not UCH1

As UCH1 and UCH2 have been found to be not associated with 26S proteasomes in Arabidopsis [31], the potential involvement of Arabidopsis RPN13 in 26S proteasome-mediated UCH1/2 activation is likely not conserved, as it is in other eukaryotes [33,34]. To investigate further, we first examined whether RPN13 is an integral subunit of Arabidopsis 26S proteasomes and RPN13’s interactions with RPN2 and UCH1/2.

According to the BoxShade and BestFit (GCG v11.1) comparisons, Arabidopsis RPN13 (14–295) shares high sequence homology with its human orthologue (23–391, 56.5% similarity and 46.0% identity), particularly in the N-terminal PRU (20–101) and C-terminal DEUBAD (178–285) domains (Appendix A). The Arabidopsis RPN13 N-terminal region (10–101) harboring the PRU is also conserved in yeast Rpn13 (3–103, 44.0% similarity and 30.8% identity), which contains only the PRU domain [22] without the C-terminal DEUBAD domain (Appendix A). 

As Rpn13 has been found in substoichiometric amounts in 26S proteasomes from humans, *D. melanogaster*, and *S. pombe* [7,25,42,43], we first examined whether RPN13 is an integral subunit of the 26S proteasomes of Arabidopsis. However, following two-dimensional gel electrophoresis and immunoblotting, RPN13 was not detected in partially purified 26S proteasomes prepared from mature (30 DAS) Col-0 rosette leaves (Appendix A). Accordingly, RPN13 was not detected using mass spectrophotometry with the single-capped 20S proteasomes (RP1-CP) collected after native gel electrophoresis was performed on partially purified 26S proteasomes from Col-0 (Table 1). 

However, the sequence-conserved RPN13 could still be transiently associated with the 26S proteasomes. As RPN13 is recruited into the 26S proteasomes by the core base subunit RPN2 in humans and budding yeast [6,11,27,28], we examined the interactions between RPN13 and two RPN2 paralogues, RPN2a and RPN2b [36], in Arabidopsis by conducting pull-down and yeast two-hybrid (Y2H) analyses. As shown in Figure 3A, *E. coli*-expressed and purified RPN13 could be readily pulled down by either GST-fused RPN2a or RPN2b. Interactions between AD-fused RPN13 and BD-fused RPN2a or RPN2b were also detected by the activation of the *HIS3* reporter in Y2H assays (Figure 3B).

According to the BoxShade (Appendix A) and BestFit (GCG v11.1) comparisons, Arabidopsis UCH1 and UCH2, the orthologues for human UCH37, are highly homologous, displaying a 67.8% identity and 77.5% similarity. A short, highly charged 12-amino-acid C-terminus is unique to UCH2 and is not found in UCH1 (Appendix A). Both UCH1 and UCH2 share high sequence homology with the human and mouse UCH37 orthologues: UCH1 (3–331) displays a 46.4%/61.2% identity/similarity with human UCH37 (8–315) and UCH2 (3–325) displays a 52.3%/67.2% identity/similarity with human UCH37 (8–320). Interestingly, short, highly charged C-terminal extensions similar to the unique 12-amino-acid UCH2 C-terminus are also present in human and mouse UCH37 (Appendix A).

We examined RPN13 interactions with UCH1 and UCH2 from Arabidopsis by conducting pull-down and Y2H analyses. Despite the high sequence similarity between UCH1 and UCH2, *E. coli*-expressed and purified RPN13 could only be readily pulled down by GST-fused UCH2 but not UCH1 (Figure 4A). In line with this, only the purified UCH2 (Figure 4B), but not UCH1 (Figure 4C), could be readily pulled down by GST-fused RPN13. Moreover, the interaction detected by *HIS3* reporter activation in the Y2H assays was only observed between AD-fused RPN13 and BD-fused UCH2 but not UCH1 (Figure 4D). 

### 2.6. Domains and Residues on Arabidopsis RPN13 and UCH2 Critical for RPN13–RPN2 and RPN13–UCH2 Interactions

As a first step in investigating the in vivo importance of RPN13–RPN2 and RPN13–UCH2 interactions, we mapped the involved critical residues and domains. The RPN2-interacting interface on the PRU domain and the potential involved residues (M31, C88, and E111) for human RPN13 were determined via NMR titration [11,28]. However, we found that the corresponding Arabidopsis RPN13 residues, M22 and E101 (M31 and E111, respectively, for hRPN13 (Appendix A)), were not involved in RPN2 binding; according to the Y2H results, AD-fused and site-specific mutagenized RPN13 (M22A or E101A) could still activate the *HIS3* reporter with BD-fused RPN2a or RPN2b, similar to wild-type RPN13 (Figure 3C, 22A and 101A vs. RPN13), indicating a potentially distinct RPN2-interacting interface on Arabidopsis RPN13. 

The ubiquitin-interacting interface on mouse RPN13 has been determined, and some of the involved critical residues, e.g., L56, F76, D79, and F98, have been determined via pull-down analyses with site-specific mutants [11]. The corresponding residues are conserved in Arabidopsis RPN13 (L47, F67, E70, and F88, respectively) and these have also been confirmed to be involved in ubiquitin binding via pull-down assays using site-specific mutants (RPN13-A47, R67, Q70, and R88, respectively) (Appendix A) [5]. Based on the NMR titration results, the RPN2-interacting interface and residues in the PRU domain of hRPN13 described above do not overlap with the ubiquitin-interacting interface [11,28]. It is expected that the conserved residues of the PRU domain in Arabidopsis RPN13, which are critical for ubiquitin binding, are not involved in RPN2 interactions. To our surprise, a GST pull-down assay showed that two of the four residues (F67 and F88) of Arabidopsis RPN13 that are critical for ubiquitin binding are also crucial for RPN2 interactions. As shown in Figure 3D, GST-fused RPN2a and RPN2b readily pulled down wild-type RPN13, RPN13-A47, and RPN13-Q70 but not RPN13-R67 or RPN13-R88. This indicates that the interaction interfaces on Arabidopsis RPN13 for ubiquitin and RPN2 partially overlap. 

The interaction between RPN13 and UCH37 is mediated by the C-terminal DEUBAD domain of RPN13 and the C-terminal UCH37-like domain (ULD) of UCH37 [29]. Using NMR and pull-down experiments, the UCH37-binding interface in the hRPN13 DEUBAD domain was mapped to two regions with confirmed residues (V286/D287/E295/I296 and K371/D373/E375/K379) [28]; these regions are rich in charged residues and complement the previously determined RPN13-binding region on UCH37 (Val313-Lys329) [25]. As some of the residues (E295, I296, K371, and E375), particularly the charged ones, are highly conserved in Arabidopsis RPN13 (E208, L209, K275, and D279, respectively; Appendix A), we expected these four conserved residues to be critical for UCH2 interactions. However, to our surprise, Y2H testing showed that, similar to AD-fused wild-type RPN13, two AD-fused dual-alanine substitution RPN13 mutants, or even a quadruple-alanine substitution mutant, could still activate the *HIS3* reporter with BD-fused UCH2 (Figure 5A,B, EL-AA, KD-AA, and ELKD-A4), suggesting that Arabidopsis RPN13 uses a distinct UCH2-interacting interface. As expected, GST pull-down assays showed that the N-terminal PRU domain, which is involved in ubiquitin and RPN2 binding, does not appear to play a role in UCH2 interactions. As shown in Appendix A, each GST-fused RPN13 variant (GST-RPN13-A47, GST-RPN13-R67, GST-RPN13-Q70, and GST-RPN13-R88), which lacks only ubiquitin binding or both ubiquitin and RPN2 binding, could still readily pull down UCH2. 

Y2H was employed to delineate the domain and residues in the C-terminus of Arabidopsis RPN13 that are critical for UCH2 interactions. AD-fused RPN13 constructs (*CΔ1*, *CΔ2*, *CΔ3*, *CΔ4*, *CΔ5*, and *CΔ6*; Figure 5A and Appendix A) encoding a series of AD-fused C-terminal deletions of RPN13 were tested individually with BD-fused UCH2 for *HIS3* reporter activation. Only AD-fused CΔ1 and CΔ2, but not CΔ3–6, activated the *HIS3* reporter (Figure 5C,D), suggesting that the region (246–269) between the truncation sites of CΔ2 and CΔ3 harbors residues critical for UCH2 interactions (Figure 5A and Appendix A). To further narrow down a smaller critical region, five AD-fused RPN13 constructs encoding all-site substitutions (generally mutated to alanine and occasionally glycine) for each of the RPN13 subregions were made within this region (246–269): A1 (246–249), A2 (250–254), A3 (255–259), A4 (261–264), and A5 (265–269) (Figure 5A,E and Appendix A). In addition, seven serial C-terminally or N-terminally combined subregion-mutated RPN13 constructs were made (*A4–5*, *A3–5*, *A2–5*, *A1–2*, *A1–3*, *A1–4*, and *A1–5*; Figure 5A,F,G and Appendix A). When tested with BD-fused UCH2, each of the AD-fused RPN13 variants with single subregion mutated (Figure 5E, A1, A2, A3, A4, and A5) or C-terminally combined subregion mutated (Figure 5F, A4–5, A3–5, and A2–5), but no N-terminally combined subregion-mutated variant (Figure 5G, A1–2, A1–3, A1–4, and A1–5), could activate the *HIS3* reporter. As all N-terminally combined subregion-mutated variants harbor A1 and A2 substitutions, this indicates that the RPN13 region 246–254 is critical for UCH2 interactions. The importance of this region for interacting with UCH2 was confirmed by conducting a GST pull-down assay; the GST-fused RPN13-A1A2 variant (the 246–254 region was substituted as indicated for the A1 and A2 subregions in Figure 5A), unlike wild-type RPN13, could not pull down UCH2 (Figure 4B, A1A2 vs. RPN13). 

Y2H was also employed to delineate the domain and residues in the C-terminus of Arabidopsis UCH2 that are critical for RPN13 interactions. Although there is high homology between UCH1 and UCH2, only UCH2 interacts with RPN13. As the highly charged C-terminal 12-amino-acid region is unique to UCH2, we suspected that the 12-amino-acid C-terminus of UCH2 is critical for interacting with RPN13. Indeed, when expressed together with *AD-RPN13*, a BD-fused *UCH2-CΔ1* construct encoding a C-terminal 13-amino-acid-truncated UCH2 fusion (the truncated sequence includes a conserved Ile that was converted to a stop codon and the unique 12-amino-acid C-terminus) could not activate the *HIS3* reporter (Figure 6A,B, UCH2 vs. UCH2-CΔ1). Interestingly, when expressed together with *AD-RPN13*, unlike BD-fused wild-type *UCH1*, a BD-fused *UCH1^2C^* construct encoding a C-terminal-swapped UCH1 fusion, in which the five C-terminal amino acids of UCH1 (AKHHP) were replaced by the unique twelve amino acids from the C-terminus of UCH2 (EKAKKQKTESST), could activate the *HIS3* reporter (Figure 6A,B and Appendix A, UCH1^2C^ vs. UCH1). The results of both the deletion and swap experiments support the importance of the unique 12-amino-acid C-terminus of UCH2 for RPN13 interactions. 

To determine the critical residues in the 12-amino-acid C-terminus of UCH2 that are involved in RPN13 interactions, we made three BD-fused *UCH2* constructs: *UCH2-A1*, *UCH2-A2*, and *UCH2-A3*, which encode BD-UCH2 fusions of all-alanine substitutions in three continuous subregions in the 12-amino-acid C-terminus of UCH2 (Figure 6A and Appendix A, A1, A2, and A3). Moreover, as a potentially larger region within the 12-amino-acid C-terminus of UCH2 could be involved in RPN13 interactions, we made additional BD-fused UCH2 constructs: *UCH2-A1–2*, *UCH2-A2–3*, *UCH2-A1/A3,* and *UCH2-A1–3*, which encode BD-UCH2 fusions of all-alanine substitutions in two or all three subregions (Figure 6A, alanine scanning variants). To our surprise, when expressed with AD-fused RPN13, all of the above BD-fused UCH2 constructs encoding all-alanine substitutions for one, two, or three subregions could still activate the *HIS3* reporter (Figure 6B–D). As no specific subregions/residues within the 12-amino-acid C-terminus of UCH2 were critical for RPN13 interactions, the 12-amino-acid C-terminal extension, regardless of the sequence identity, is likely not directly involved in forming an RPN13-interacting interface. Instead, it is perhaps involved in stabilizing the structural integrity of an upstream RPN13-interacting fold/interface. In line with this, when expressed with AD-fused RPN13, BD-fused *UCH1^2C-A2^* and *UCH1^2C-A3^*, encoding BD-UCH1 with a swapped 12-amino-acid UCH2 C-terminus harboring all-alanine substitutions for the A2 or A3 subregion, just like BD-*UCH1^2C^*, could also activate the *HIS3* reporter (Figure 6A,C). Further experiments are needed to delineate the UCH2 residues/domains directly involved in RPN13 interactions.

### 2.7. F67 Residue of RPN13, Critical for Both RPN2 and Ubiquitin Binding, Is Important In Vivo 

A rational approach to determining the in vivo importance of the discovered RPN13 domain and residues critical for interacting with ubiquitin [5], as well as RPN2 and UCH2, is to perform functional complementation of the null mutant *rpn13-1* using site-specific mutated RPN13 variants. However, in agreement with our previous reports [10,19], no clearly discernible vegetative or reproductive growth defects could be observed with *rpn13-1*; an alternative in vivo assessment approach was thus required. Moreover, wild-type-like growth phenotypes were also observed with the double null mutant *rpn13-1 rpn10-2* harboring *RPN10-u123*, a mutant line affecting both RPN10- and RPN13-mediated recognition functions for ubiquitinated substrates [19], which also prohibits functional complementation in this line. Fortunately, we observed a complete loss of segregation of homozygous *rpn10-2* from heterozygous *rpn10-2* plants in a homozygous *rpn13-1* background [19], which allowed for functional assessments of the various critical RPN13 domains and residues involved in ubiquitin, RPN2, and UCH2 interactions.

It would be interesting to determine the underlying role of RPN13 in conferring the residual 2% segregation rate of homozygous *rpn10-2* from heterozygous *rpn10-2* plants and the mechanism. To examine whether the residues/domain involved in ubiquitin, RPN2, or UCH2 interactions are critical for the 2% homozygous *rpn10-2* segregation rate, we independently introduced a construct encoding wild-type RPN13 or each of the RPN13 mutants (RPN13-R67, RPN13-Q70, and RPN13-A1A2, which are defective in interacting with ubiquitin/RPN2, ubiquitin, and UCH2, respectively) into heterozygous *rpn10-2* with a homozygous *rpn13-1* background. We expected that wild-type RPN13 could restore the 2% segregation rate of homozygous *rpn10-2*. When homozygous *rpn10-2* was segregated from heterozygous *rpn10-2* in a homozygous *rpn13-1* background harboring RPN13-R67, RPN13-Q70, or RPN13-A1A2, the recovery levels of the 2% segregation rate could potentially reflect the relative importance of RPN13 in binding to RPN2, ubiquitin, and UCH2, respectively.

Heterozygous *rpn10-2* plants with a homozygous *rpn13-1* background (*RPN10rpn10*/*rpn13rpn13*) harboring wild-type *RPN13* (*N10n10*/*n13n13* [*N13*]), *RPN13-R67* (*N10n10*/*n13n13* [*N13-R67*]), *RPN13-Q70* (*N10n10*/*n13n13* [*N13-Q70*]), or *RPN13-A1A2* (*N10n10*/*n13n13* [*N13-A1A2*]) were generated for segregation analyses. However, the expression of RPN13-A1A2 proteins was barely detected in all *N10n10*/*n13n13* [*N13-A1A2*] plants despite repeated efforts from multiple independent lines. Homozygous *rpn10-2* segregation was thus not conducted with *N10n10*/*n13n13* [*N13-A1A2*] plants because any significantly reduced segregation rate could be attributed to the extremely low expression levels of this RPN13 variant. Two independent lines for *RPN10rpn10*/*rpn13rpn13* harboring wild-type *RPN13* (*N10n10*/*n13n13* [*N13*]), *RPN13-R67* (*N10n10*/*n13n13* [*N13-R67*]), or *RPN13-Q70* (*N10n10*/*n13n13* [*N13-Q70*]) were selected for segregation analyses. Before this, *RPN10rpn10*/*rpn13rpn13* harboring wild-type and site-specific mutated *RPN13* constructs were confirmed by PCR for *rpn10-2* heterozygosity and *rpn13-1* homozygosity (Appendix A). Moreover, the RPN13 variant expression levels in each of the two selected lines harboring *RPN13-R67* (*R67#1*–*2*) or *RPN13-Q70* (*Q70#1*–*2*) were similar to those in the two lines harboring wild-type *RPN13* (*N13#1*–*2*) (Appendix A). The expression levels of wild-type and site-specific mutated RPN13 variants from these lines were also at similar levels to wild-type RPN13 in Col-0 and the heterozygous *rpn10-2* plants with a Col-0 background (Appendix A, left two lanes).

In line with previous reports [10,19], while a 1.93% (69/3610) rate for homozygous *rpn10-2* segregation from heterozygous *rpn10-2* plants with a Col-0 background was observed (Table 2, *N10n10*/*N13N13*), no (0/1348) homozygous *rpn10-2* was segregated from heterozygous *rpn10-2* plants with an *rpn13-1* null mutant background (Table 2, *N10n10*/*n13n13*). As expected, when wild-type *RPN13* was introduced into heterozygous *rpn10-2* plants with a homozygous *rpn13-1* background (*N10n10*/*n13n13* [*N13-1*] and *N10n10*/*n13n13* [*N13-2*]), a segregation rate of nearly 2% was restored (Table 2, 2.04% (57/2825) and 1.78% (59/3330), respectively). Significantly, when the construct encoding RPN13 variant R67 was introduced into heterozygous *rpn10-2* plants with a homozygous *rpn13-1* background (*N10n10*/*n13n13* [*R67-1*] and *N10n10*/*n13n13* [*R67-2*]), this rate was only partially restored (Table 2, 1.10% (34/3155) and 0.84% (26/3111), respectively). In contrast, when the RPN13 variant Q70 was introduced into the heterozygous *rpn10-2* plants with a homozygous *rpn13-1* background (*N10n10*/*n13n13* [*Q70-1*] and *N10n10*/*n13n13* [*Q70-2*]), the 2% homozygous *rpn10-2* segregation rate was apparently restored (Table 2, 2.48% (77/3124) and 2.25% (66/2955), respectively). Only the RPN13 variant R67, which is defective in terms of both RPN2 and ubiquitin binding, but not Q70, which is only defective in ubiquitin binding (Figure 3D and [5]), could not fully restore the 2% rate, suggesting that the RPN2-binding activity and not the ubiquitin-binding activity of RPN13 is critical for the 2% rate of homozygous *rpn10-2* segregation from heterozygous *rpn10-2* plants.

## 3. Discussion

It is well known that several classes of evolutionarily conserved ubiquitin receptors, including the intrinsic proteasome subunits RPN1, RPN10, and RPN13, and extrinsic UBL-UBA shuttle receptors, play a pivotal role in targeting ubiquitinated cellular regulators for proteasome turnover [1,4]. Although extensive biochemical and functional studies have been carried out on these major ubiquitin receptors (reviewed in [4]), the accumulated evidence does not sufficiently or unequivocally confirm their in vivo importance for ubiquitin-binding activities [10,19]. Moreover, confirming their importance is complicated by the presence of additional domains harboring distinct biochemical activities in these major proteasome ubiquitin receptors. For example, the N-terminal vWA domain of RPN10 and the C-terminal DEUBAD domain of RPN13 are involved in lid–base association [14] and UCH37 recruitment [25,26,27], respectively. In this study, we further confirmed that the ubiquitin-binding activities of Arabidopsis RPN10 are dispensable in vivo, as the C-terminal truncated RPN10 variant N215, with the UIM1–3 motifs deleted, could still rescue all the vegetative and reproductive growth defects associated with the *RPN10* null mutant *rpn10-2*. In contrast, the structural role of RPN10 in the 26S proteasome involving the N-terminal vWA domain is likely more essential for its functions in vivo. 

In this study, we also observed Arabidopsis RPN13’s interactions with RPN2 and UCH2, in which a novel RPN2-interacting interface on RPN13, which partial overlaps with ubiquitin-binding residues located in the PRU domain, and a C-terminal region (246–254) involved in UCH2 interactions were discovered. The observed RPN13 association with RPN2 and UCH2 suggests the possible role of RPN13, via RPN2, in UCH2 recruitment and 26S proteasome association. Interestingly, the residual homozygous *rpn10-2* segregation was partially compromised by a site-specific mutated RPN13 variant RPN13-F67R, which abolished RPN2 and ubiquitin binding, but not RPN13-E70Q. The latter only abolished ubiquitin binding, suggesting that the critical role played by RPN13 in the residual homozygous *rpn10-2* segregation could be mediated, at least partially, by RPN2 association.

### 3.1. The Ubiquitin-Binding Motifs UIM1–3 of RPN10 Are Dispensable In Vivo

The importance of ubiquitin-binding activities for RPN10 in vivo has often been proposed in studies examining deletion mutants in various species [16,17,18]. Using this reasoning, Smalle et al. suggested that the growth defects associated with a T-DNA-inserted mutant *rpn10-1*, which expresses a C-terminal UIM1–3-truncated RPN10 (1–186) fused to NPT-II, could primarily be attributed to deficient recognition of ubiquitinated proteins [21]. However, our previous report claimed that the ubiquitin-binding motifs UIM1-3 are dispensable because an RPN10 variant, RPN10-u123, in which each ubiquitin-binding motif (UIM1–3) was mutated at multiple sites, could complement all the growth defects associated with a different and phenotypically more severe T-DNA-inserted null mutant (*rpn10-2*) [10]. For the *rpn10-1* mutant, although low levels of a truncated RPN10 fusion protein could still be assembled into the 26S proteasome [21], the truncation point occurred very close to the edge of the vWA domain (1–189), potentially causing vWA destabilization. To investigate this further, we complemented *rpn10-2* with a C-terminally truncated RPN10 variant (1–215), N215, which covers the full vWA domain but deletes all the UIM motifs (221–324). Although N215 expression could not be detected in all the *N215*-complemented *rpn10-2* lines, all the examined growth defects of *rpn10-2* were, surprisingly, completely rescued (Figure 1 and Appendix A). Our results again confirmed that the UIM1–3 ubiquitin-binding motifs are dispensable in terms of the RPN10-associated functions observed with *rpn10* mutants. In addition, the N-terminal region (N215) harboring the vWA domain appears to be fully functional in vivo, as all growth defects associated with *rpn10-2* plants were completely rescued.

### 3.2. The N-Terminal Region of RPN10 Harboring the vWA Domain Is Functionally Important In Vivo 

As an initial effort to further delineate the N-terminal region of RPN10, a functional assessment was conducted by complementing *rpn10-2* defects with various *RPN10* constructs encoding small N-terminal deletion and single- or multiple-substitution variants, including *NΔ15*, *NΔ42*, *NΔ53*, *11A*, *11R*, *11K*, *11*–*12RR*, and *10*–*14A5*. Among these eight tested constructs, the T3 lines, which could partially rescue the *rpn10-2* defects and are double homozygous for the *rpn10-2* allele and a specific *RPN10* N-terminal variant, could only be generated with *11A* but not with any of the remaining constructs. The D11 residue in a DXS motif is reportedly involved in a critical intra-domain salt bridge that stabilizes the vWA domain of RPN10 [14], similar to the role played by the D520 residue in the DXS motif of a vWA domain in von Willebrand factor, as described previously [35]. Except for the *11A* construct, the T3 homozygous lines obtained for the rest of N-terminal constructs generally associated with heterozygous *rpn10-2* and the expression of any particular N-terminal variant was not detected. While T3 homozygous lines such as *10*–*14A5* were found to be associated with homozygous *rpn10-2*, growth defects for *rpn10-2* were not rescued and 10–14A5 protein expression was also not detected. It is conceivable that because the neutral substitution (D11A) could only partially rescue the growth defects of *rpn10-2*, other non-conservative substitutions (11R, 11K, 11–12RR, and 10–14A5) or small N-terminal deletions (*NΔ15*, *NΔ42*, and *NΔ53*) surrounding D11 likely caused disruption and destabilization of the vWA fold, which made these N-terminal variants highly unstable and degraded, compromising the critical in vivo activity of RPN10. These results suggest that the N-terminal region of Arabidopsis RPN10 that harbors a vWA domain is sensitive to manipulation by small deletions or site-specific substitutions. In line with this, the C-terminal UIM1–3-deleted RPN10 (1–186) fusion in *rpn10-1*, which is truncated near the edge of the vWA domain, was found to be expressed at an extremely low level and, although still integrated into 26S proteasomes, was functionally compromised [21]. 

Most strikingly, RPN10-A11 was not found in the 26S proteasomes prepared from *A11*-compemented *rpn10-2*, suggesting a critical role of D11 in RPN10 assembly into 26S proteasomes. However, although the absence of RPN10 in the 26S proteasomes in *rpn10-2* was associated with the lowest double-capped 20S proteasome abundance and strong growth defects, a lack of RPN10-11A in the 26S proteasomes in *11A*-complemented *rpn10-2* was only associated with intermediate levels of double-capped 20S proteasomes and intermediate growth defects. These results suggest that *rpn10-2* defects, if associated with the altered structure/activity of 26S proteasomes, are likely not simply due to the absence of RPN10. However, undetectable levels of N215 fully rescued the *rpn10-2* defects, suggesting that undetectable or extremely low levels of functional RPN10 are sufficient in vivo. Thus, we cannot completely rule out that extremely low levels of N215 and RPN10-11A are still assembled into 26S proteasomes.

### 3.3. Conformational Changes or Structural Defects Are Likely Associated with 26S Proteasomes in rpn10-2

Several lines of evidence suggest that conformation changes or structural defects, which could be responsible for most or part of the observed growth defects, are likely associated with 26S proteasomes in *rpn10-2*. First, comparing the abundance of double-capped 20S proteasomes in Col-0, as well as wild-type *RPN10*-, *u123*-, and *N215*-complemented *rpn10-2*, which all showed wild-type growth phenotypes, to those of *A11*-complemented *rpn10-2* and *rpn10-2*, which showed intermediate or severe growth defects, we observed a general correlation between reduced levels of double-capped 20S proteasomes and growth defect severity. This suggests that the possible conformational changes in single-capped 20S proteasomes prohibits their further assembly into double-capped 20S proteasomes. It is known that CP-RP association is mediated by C-terminal HbYX motif docking of several RPT subunits into pockets between adjacent CP alpha-subunits (reviewed by [39]); this could be modulated by specific interactions between the lid subunits RPN5 and RPN6 and alpha ring subunits α1 and α2 [44,45]. It is conceivable that the CP-RP association of 26S proteasomes in *rpn10-2* could potentially be altered.

Second, besides the lack of RPN10, the results of the mass spectrometry analyses for single-capped 20S proteasomes collected after PEG purification and native gel electrophoresis suggest that a portion of the single-capped 20S proteasomes in *rpn10-2*, in comparison with those from Col-0, are likely composed of specific paralogues for several core subunits. Based on the normalized total number of PSMs, the CP subunits PAB2, PAF2, PBB2, PBC2, and PBE2, base subunit RPT6b, and lid subunit RPN8b were found in single-capped 20S proteasomes from *rpn10-2* but not in those from Col-0 (Table 1). Also, the lid subunit RPN9b (2.3-fold) was significantly more abundant in single-capped 20S proteasomes from *rpn10-2* than in those from Col-0. In contrast, the RPN9b paralogue RPN9a was only observed in single-capped 20S proteasomes from Col-0 but not in those from *rpn10-2*. The appearance of specific paralogues for various subunits in a portion of single-capped 20S proteasomes from *rpn10-2* could be due to low expression levels of the corresponding encoding genes in Col-0 and drastic activation in *rpn10-2* (Appendix A) by coordinated transcriptional regulation induced by proteasome stresses or mutations [39]. Because of the high sequence homology between paralogues for the various unique subunits in a portion of the *rpn10-2* 26S proteasomes (in the range of 91.3–97.8/93.5–98.5 identity/similarity), it is conceivable that the structural alteration caused by the presence of specific paralogues of various core subunits in *rpn10-2* 26S proteasomes could be subtle. Interestingly, the *rpn10-2*-specific paralogue PBE1 and the Col-0-specific paralogue RPN9a have a 24 a.a. insertion and a 63 a.a. deletion, respectively, in comparison with their corresponding paralogues. Notably, although the six residues (T58, D74, K114, S211, D248, and S251) involved in forming the putative catalytic center [46] for PBE1 are conserved, the unique 24 a.a. insertion in PBE1 is located between residues D74 and K114 within the region forming the catalytic center. The prominent lack of an RPN10 subunit, together with the possible subtle structural alteration caused by the presence of specific paralogous for various core subunits, could potentially cause major conformational changes or structural defects in *rpn10-2* 26S proteasomes. However, because RPN10-11A was also absent in the 26S proteasomes isolated from *11A*-complemented *rpn10-2*, and because it was only associated with intermediate growth defects, the severe *rpn10-2* defects cannot be entirely attributed to a lack of RPN10 in the 26S proteasomes. Nevertheless, as described above, we cannot completely rule out that extremely low levels of RPN10-11A were still assembled into the 26S proteasomes. This could potentially provide partial complementation of the growth defects and abundance of double-capped 20S proteasomes associated with *rpn10*-2.

Potential conformational changes or structural defects in *rpn10-2* 26S proteasomes were also indicated by the increase in associated regulatory proteins involved in quality control, assembly, and/or disassembly. Most interestingly, two well-known 26S proteasome-associated regulatory proteins, PA200 (Blm10 in yeast) and ECM29, were found to be increased in abundance, suggesting an enhanced association with the single-capped 20S proteasomes from *rpn10-2* (Table 1). PA200 is generally considered to play a regulatory role in CPs and serves as an alternative activator of 20S proteasomes [37,47,48]. The increased abundance of PA200 could simply be due to the drastically induced expression of 20S proteasomes, as described here (Figure 1B) and previously [10,36] in *rpn10* mutants. The increased abundance of PA200 in *rpn10-2* proteasomes also indicates that potential proteotoxic stress is associated with *rpn10-2*, as was suggested in previous reports [37,47,48]. PA200-CPs could potentially be involved in the degradation of ATP- and ubiquitin-independent proteasome substrates. Alternatively, it could be involved in CP assembly [49], stability [50,51], substrate entry [50,51], or subcellular localization [52].

In particular, the increased abundance of ECM29 in single-capped 20S proteasomes from *rpn10-2* supports the idea that there are conformational changes or structural defects in the 26S proteasomes in *rpn10-2*. Initially identified in yeast and mammals as a salt-sensitive proteasome component with the capacity to stabilize proteasomes in the absence of ATP, ECM29 has been proposed to function as an RP-CP tethering factor [53,54]. Intriguingly, an enhanced association between ECM29 and structurally aberrant 26S proteasomes, such as those with defective CP maturation [55] or CP-RP interfaces [41,56], has often been found. ECM29 inhibits aberrant 26S proteasomes by compromising ATPase activity and gate opening [41,57], appearing to play a critical role in quality control checkpoints for 26S proteasomes. In addition, the role of ECM29 in 26S proteasome disassembly under oxidative stress and glucose starvation was also discovered [58,59]. The increased abundance of ECM29 could represent an enhanced association due to the structural aberrance associated with single-capped 20S proteasomes to impose a quality control checkpoint. In line with its role in disassembly under stress [58,59], an enhanced single-capped 20S proteasome–ECM29 association could inhibit further CP attachment, thus explaining the reduced abundance of double-capped 20S proteasomes in *rpn10-2*. Interestingly, structural modeling has indicated direct interactions between ECM29 and RPN10 in the assembled state [59] and suggested that ECM29 intrudes on CP-RP interactions by contacting several base subunits, including RPT1, RPT4, RPT5, RPN1, and RPN10, thus disassembling 26S proteasomes under oxidative stress [58]. Further investigation is required to determine whether RPN10 is directly or indirectly involved in ECM29’s association with 26S proteasomes. 

All the evidence provided above supports the idea that there are conformational changes or structural defects in *rpn10-2* 26S proteasomes. Further biochemical experiments are required to directly measure readouts and characterize the nature of the structural alteration of *rpn10-2* 26S proteasomes. These experiments include an in vitro activity assay using standard or specific ubiquitinated substrates or cryo-EM structural analyses with purified *rpn10-2* 26S proteasomes.

### 3.4. RPN13’s Role in UCH2 Recruitment to 26S Proteasomes by RPN2 Could Be Relevant In Vivo 

Besides being established as an intrinsic ubiquitin receptor [6,11], human RPN13 (hRPN13) was originally found to play a major role in the recruitment and activation of a dominant 26S proteasome-associated DUB UCH37, modulating the turnover of ubiquitinated substrates [25,26,27]. The proteasome recruitment and activation of UCH37 by hRPN13 is mediated by hRPN13’s attachment to the core base subunit RPN2 [6,11,27,28]. However, two Arabidopsis orthologues of human UCH37, UCH1 and UCH2, were found to not be associated with 26S proteasomes prepared under mild conditions [31]. Moreover, although RPN13 has been found to be present in substoichiometric amounts in 26S proteasomes from humans, *D. melanogaster*, and *S. pombe* [7,25,42,43], Arabidopsis RPN13 was not found in 26S proteasomes prepared by PEG precipitation (in this study) or using affinity-purification methods [37,38]. Nevertheless, it is conceivable that the RPN2-mediated hUCH37 orthologue recruitment by RPN13 could be more transient in Arabidopsis. Alternatively, interactions between RPN13 and RPN2 and/or hUCH37 orthologues may not be sufficiently stable to be detected when using the various preparation methods for Arabidopsis 26S proteasomes [31,37,38].

To explore the potential role of RPN13 in RPN2-mediated hUCH37 orthologue recruitment in Arabidopsis, we thoroughly examined its interactions with RPN2 paralogues (RPN2a/2b), as well as with UCH1/UCH2, and delineated the critical residues and domains involved in these interactions. Interactions between RPN13 and both RPN2 paralogues were confirmed by GST pull-down and Y2H assays (Figure 3). Interestingly, RPN13 was found to only interact with one hUCH37 orthologue, UCH2, but not with UCH1 (Figure 4).

To investigate further, the domains and residues of Arabidopsis RPN13 and UCH2 that are critical for RPN13–RPN2 and RPN13–UCH2 interactions were expertly delineated. Critical RPN2-binding residues in the PRU domain of human hRPN13 (M31, C88, and E111) have been determined by NMR titration [11,28]; using Y2H, two corresponding conserved residues in Arabidopsis RPN13, M22 and E101 (for hRPN13 M31 and E111, respectively), were found to not be involved in RPN2a and RPN2b interactions. Interestingly, while the RPN2-binding interface in the PRU domain of human hRPN13 was found to be independent of the interface involved in ubiquitin binding [11,28], we surprisingly observed that some conserved residues critical for ubiquitin binding, including F67 and F88 but not L47 and E79, were also critical for RPN2 binding. These results suggest that the interaction interfaces in the PRU domain of Arabidopsis RPN13 for ubiquitin and RPN2 partial overlap.

Similarly, although some confirmed residues of the UCH37-interacting regions in the C-terminal DEUBAD domain of hRPN13 [28] are conserved in Arabidopsis RPN13 (E295, I296, K371, and E375), when tested as dual or quadruple site-specific mutations using Y2H, none of these conserved residues were critical for UCH2 binding, suggesting that Arabidopsis RPN13 uses a unique UCH2-interacting interface. After carefully examining deletion and alanine scanning mutations in the sub-regions, the region of RPN13 (246–254) critical for UCH2 interaction was mapped using Y2H and verified by pull-down assays. This region is located between the two mapped regions of hRPN13 within the DEUBAD domain that are critical for UCH37 binding [28]. Moreover, the UCH2 interface that is critical for RPN13 binding was also investigated. In line with the assertion that the human UCH37 C-terminus (Val313–Lys329) is critical for hRPN13 binding [25], we verified, via deletion and swap experiments, that the C-terminus (Ile218–Thr330) of Arabidopsis UCH2 is also critical for RPN13 interactions. However, all-alanine scanning mutants for one, two, and three subregions of the UCH2 C-terminus (Glu219–Thr330) could not disrupt UCH2–RPN13 interactions, suggesting that the C-terminus of UCH2, regardless of the sequence identity, likely plays a role in stabilizing an upstream RPN13-interacting interface. The exact UCH2 interface directly involved in RPN13 binding requires further elucidation.

To explore the potential role of RPN13 in RPN2-mediated UCH2 recruitment to 26S proteasomes in Arabidopsis, the in vivo importance of critical interfaces involved in RPN13–RPN2 and RPN13–UCH2 interactions must be validated. Because no clearly discernible growth defects could be observed with the *RPN13* null mutant *rpn13-1*, an alternative in vivo assessment approach was required. Fortunately, the complete loss of the 2% rate of homozygous *rpn10-2* segregation from heterozygous *rpn10-2* plants in a homozygous *rpn13-1* background [19] allowed us to determine the functional importance of the various critical RPN13 domains and residues involved in interacting with ubiquitin, RPN2, and/or UCH2. We segregated homozygous *rpn10-2* from heterozygous *rpn10-2* in a homozygous *rpn13-1* background harboring wild-type RPN13 or various mutants. RPN13-R67, RPN13-Q70, and RPN13-A1A2 are defective in terms of their interaction with ubiquitin/RPN2, ubiquitin, and UCH2, respectively. Meaningful segregation results could not be achieved for those harboring RPN13-A1A2, as RPN13-A1A2 protein expression was barely detected. Nevertheless, RPN13–RPN2 interactions appeared to be important in vivo. Although the complete restoration of the 2% homozygous *rpn10-2* segregation rate was observed in those harboring wild-type RPN13 or RPN13-Q70 (defective only in ubiquitin binding), those harboring the RPN13 variant R67, which is defective in both RPN2 and ubiquitin binding, had a significantly reduced rate (only 0.8 or 1.1% obtained for two independent lines). These results support the idea that the RPN2-binding and not the ubiquitin-binding activity of RPN13 is at least somewhat critical in maintaining this 2% segregation rate. 

The complete loss of the residual segregation of homozygous *rpn10-2* in an *rpn13-1* background for heterozygous *rpn10-2* plants (Table 2 and [19]) supports the claim that RPN13 is involved in the residual 2% segregation. The role of RPN13 could, in part, be mediated through its association with RPN2, because a defective RPN2-binding residue on RPN13 significantly dampened the residual 2% segregation of homozygous *rpn10-2*. Our results suggest that a potentially transient recruitment of RPN13 to the 26S proteasomes through RPN2 could be functionally important. Together with the observed interactions between RPN13 and RPN2 and between RPN13 and UCH2, the potential importance of RPN2-mediated UCH2 recruitment to the 26S proteasome by RPN13 in Arabidopsis warrants further interrogation. 

## 4. Materials and Methods

### 4.1. Plant Materials and Growth Conditions

The source of the *rpn10-2* and *rpn13-1* lines has been described previously [10]. Heterozygous *rpn10-2* plants with a homozygous *rpn13-1* background were generated by crossing *rpn13-1* into heterozygous *rpn10-2* plants. To grow the Arabidopsis plants, the seeds were surface-sterilized using 20% bleach and 0.1% Tween 20 and stratified on half-strength Murashige and Skoog plates (0.8% agar, pH 5.8) supplemented with 1% sucrose at 4 ℃ for 3 d in the dark. The seeds were then germinated in a growth chamber at 22 °C (16 h light/8 h dark). Nine-day-old seedlings were transferred to soil (a 6:1:1 mixture of humus/vermiculite/perlite) and grown using a 16 h light/8 h dark photoperiod with a light intensity of ~120 mmol·m^−2^·s^−1^ at 22 °C. For dark-induced leaf senescence, 30 DAS rosette leaves were detached, placed on Petri dishes containing two layers of 3MM chromatography paper (Whatman, Maidstone, UK) soaked in sterile water, and incubated in the dark at 22 °C. To perform crosses, the sepals, petals, and stamens were removed from stage-12 flowers [60] two days before hand pollination of the pistils with anthers from the male parent. Homozygous *rpn10-2* plants were identified at the early seedling stage from the progeny of selfed heterozygous plants by their reduced growth rate and leaf morphology and, subsequently, by genotyping.

### 4.2. Complementation and Segregation Analyses of rpn10-2

To test the complementation of *rpn10-2* defects, we constructed the following binary plasmids for various deletion and N-terminal site-specific mutants of RPN10: ptRPN10-NΔ15, ptRPN10-NΔ42, ptRPN10-NΔ53, ptRPN10-N215, ptRPN10-11A, ptRPN10-11K, ptRPN10-11R, ptRPN10-11–12RR, and ptRPN10-10–14A5, which are described in Appendix A. To generate segregation lines for *RPN10 rpn10 rpn13 rpn13* harboring each RPN13 variant, we constructed the following binary plasmids: ptRPN13, ptRPN13-R67, ptRPN13-Q70, and ptRPN13-A1A2, which are also described in Appendix A. Each of these binary plasmids was separately transformed into *Agrobacterium tumefaciens* GV3101 using a freeze–thaw method and then transformed into heterozygous *rpn10-2* with a Col-0 or *rpn13*-*1* background using a floral dip method [61]. To test the complementation of *rpn10-2* defects, the heterozygous *rpn10-2* plants were transformed independently with each of the above-described *RPN10* constructs. T1 seedlings containing a complementation construct and *rpn10-2* were first selected according to kanamycin and sulfadiazine resistance. T3 plants, homozygous for each of the RPN10 variants containing the *rpn10-2* allele, were generated by kanamycin segregation with T3 plants. Segregation was performed by genotyping with T3 or T4 plants homozygous for each of the RPN10 variants containing the *rpn10-2* allele to obtain plants homozygous for the *rpn10-2* allele and each RPN10 variant. 

To generate segregation lines for *RPN10 rpn10 rpn13 rpn13* harboring each RPN13 variant, the *RPN10 rpn10 rpn13 rpn13* plants were transformed independently with each of the above-described *RPN13* constructs. For each construct, 50 out of 120 kanamycin-resistant (Kan+) T1 plants were genotyped to collect heterozygous *rpn10-2*/Kan+ plants. The protein expression levels of the RPN13 variants were then analyzed for 15–20 independent T1 lines of *RPN10/rpn10 rpn13/rpn13* harboring wild-type RPN13, RPN13-R67, RPN13-Q70, or RPN13-A1A2. Apart from RPN13-A1A2, which displayed extremely low expression levels, independent T1 lines harboring each RPN13 variant with expression levels similar to those of wild-type RPN13 in Col-0 and the heterozygous *rpn10-2* plants with a Col-0 background were selected to establish multiple T3 lines of *RPN10/rpn10 rpn13/rpn13* harboring each variant in homozygosity for the *rpn10-2* segregation analyses. 

### 4.3. Isolation and Analyses of Proteasome Complexes

The proteasome complexes from 1 g of rosette leaves of 30–36 DAS plants were partially purified with 10% (*w*/*v*) polyethylene glycol (PEG) 8000 (Sigma, St. Louis, MO, USA) at 4 °C, as described previously [36]. To visualize the abundance of single- and double-capped 20S proteasomes, and RPN10’s integration into these complexes for various complemented *rpn10-2* lines, proteasome complexes were separated using 4% native PAGE at 4 °C and detected via an in-gel activity assay using Suc-LLVY-AMC (Enzo Life Sciences, Plymouth Meeting, PA, USA), as described previously [10], or via immunoblotting using antisera against RPT5 (Enzo Life Sciences, Plymouth Meeting, PA, USA) or RPN10 (custom-made by Genesis Biotech, Taipei, Taiwan). The in-gel proteasome activity was visualized using a built-in UV illuminator on a BioSpectrum 600 (UVP, Upland, CA, USA). To examine the association of RPN13 with proteasome complexes, the latter were separated using 4% native PAGE and further separated via second-dimensional SDS-PAGE following immunoblotting using rabbit polyclonal antisera against recombinant Arabidopsis RPN13 (custom-made by Genesis Biotech, Taipei, Taiwan) or moss 20S proteasomes (a kind gift from Dr. Pirre-Alain Girod). 

### 4.4. In-Gel Trypsin Digestion and Mass Spectrometry Analysis for Protein Identification and Analysis

To identify unique peptides for each 26S proteasomes subunit, single-capped 20S proteasomes were collected after separation using 4% native PAGE for PEG-precipitated 26S proteasomes of equal amounts from *Col-0* and *rpn10-2* (three biological repeats for each genotype). These were then subjected to Tandem Mass Spectrometry Analysis. The collected gel slices containing single-capped 20S proteasomes were trypsin-digested, as described previously [62]. After trypsin digestion, the dried pellet was dissolved in 0.1% formic acid for LC-MS/MS analysis. An LTQ XL mass spectrometer model (Thermo Fisher Scientific, Waltham, MA, USA), coupled with an on-line ultra-performance liquid chromatography system (UPLC, nanoACQUITY, Waters Corp., Milford, MA, USA), was utilized for protein identification and analysis. A Symmetry C18 trap (180 µm × 2.0 cm, 5 µm, Waters Corp.) and a BEH130 C18 nanoACQUITY UPLC column (75 µm × 10 cm, 1.7 µm, Waters Corp.) were used to deliver the solvent and separate tryptic peptides using a linear gradient (from 5% to 40%) of acetonitrile in 0.1% (*v*/*v*) formic acid for 90 min at a flow rate of 300 nL/min. MS data were acquired via a full MS scan followed by four MS/MS scans of the top four precursor ions. The MS scan was performed over the mass-to-charge (*m*/*z*) range of 400 to 1800 using the data-dependent acquisition mode with dynamic exclusion enabled. Peptide identification was performed, using the Proteome Discoverer software (v1.3, Thermo Fisher Scientific) with SEQUEST and Mascot (v2.3, Matrix Science Inc., Boston, MA, USA) search engines, against the TAIR10 database with 27,241 protein entries downloaded from the Arabidopsis Information Resource website (http://www.arabidopsis.org/, accessed on 7 November 2012). The parameters for the database searches were set as follows: full trypsin digestion with 2 maximum missed cleavage sites; precursor mass tolerance: 2 Da; fragment mass tolerance: 1 Da; dynamic modifications: oxidation (M); and static modifications: carbamidomethyl (C). The identified peptides were validated, using the Percolator algorithm, against a decoy database search, which rescored the peptide spectrum matches (PSM) using *q*-values and posterior error probabilities. All the peptides were filtered with a *q*-value threshold of 0.01 (1% false discovery rate) and proteins were filtered with a minimum of 2 peptides per protein; only the top-ranked peptides and peptides of top-scoring proteins were counted. For all proteins identified from single-capped 20S proteasomes, the abundance was quantified as the total number of PSMs [63]. The total number of PSMs was based on subunit-specific peptides that were detected and normalized according to PBF1 (a subunit with a single member) of the core particle. If common peptides for paralogues were present, their average number of PSMs were partitioned according to the ratio of the number of PSMs for subunit-specific peptides before normalization.

### 4.5. RNA-seq Analysis

For the RNA-seq analysis, the total RNA from 21 DAS rosette leaves of Col-0 and *rpn10-2* plants (three duplicates each) were extracted using an RNeasy Plant Mini Kit (QIAGEN, Hilden, Germany). Libraries for RNA-seq were prepared using the Illumina TruSeq stranded mRNA sample preparation kit (Illumina, San Diego, CA, USA) according to the manufacturer’s protocol. Briefly, 2 μg of total RNA was used for the construction of each library. PolyA RNA was captured by oligodT beads and fragmented after they were eluted. The first-strand cDNAs were then synthesized by reverse transcriptase (SuperScript III, 18080-093, Invitrogen, Carlsbad, CA, USA) using dNTP and random primers. The second-strand cDNAs were generated using a dUTP mix. The double-stranded cDNAs were subjected to the addition of a single ‘A’ base to the 3′ end, followed by barcoded Truseq adapter ligation. Finally, the products were purified and enriched using 12 cycles of PCR to create the final double-stranded cDNA library. To check the quality, we used the BioRad QX200 Droplet Digital PCR EvaGreen supermix system (Cat. No. #1864034; BioRad, Hercules, CA, USA) and Fragment Analyzer 5200 (Agilent, Santa Clara, CA, USA) with an HS NGS fragment (1–6000 bp) kit to estimate the quantity and check the size distribution. The prepared libraries were pooled for paired-end sequencing using an Illumina Hiseq 4000 at Yourgene Health Co. (New Taipei City, Taiwan) with 150 bp paired-ended sequence reads. 

RNA-seq reads were first processed using Trimmomatic [64] to remove adapters and reads shorter than 80 bps. Clean reads were mapped to the transcriptome database of Araport11 [65] using Bowtie2 [66] and only alignments of paired reads which mapped to the same transcripts were accepted. The rest of the reads were mapped to the TAIR10 genome using BLAT [67]. All accepted alignments with identical bases greater than 95% of the read length and at least 97% of reads from each RNA-seq library were mapped. Gene read counts were computed using the RackJ toolkit (https://rackj.sourceforge.net/, accessed on 21 October 2021) and normalized into log-count-per-million values using the TMM method [68]. Normalized RPKM values were then derived based on the log-count-per-million values.

### 4.6. GST Pull-Down Analyses

The purification of GST-fused and His-/T7-tagged recombinant bait and prey proteins and the GST pull-down assays were conducted as described previously [5]. The pull-down products were boiled for 5 min in sample buffer, separated by SDS-PAGE, and analyzed via immunoblotting and chemiluminescence (Perkin–Elmer, Shelton, CT, USA) using mouse anti-T7 serum (Novagen, Darmstadt, Germany) and horseradish peroxidase-conjugated goat anti-mouse IgG serum (Santa Cruz Biotechnology, Dallas, TX, USA) as the primary and secondary antibodies, respectively. The constructs made for the GST-fused baits are described in Appendix A. These include the following: GST-RPN2a, GST-RPN2b, GST-UCH1, GST-UCH2, GST-RPN13, GST-RPN13-A47, GST-RPN13-R67, GST-RPN13-Q70, GST-RPN13-R88, and GST-RPN13-A1A2. The constructs made for the His-/T7-tagged preys are also described in Appendix A and include the following: RPN13, RPN13-A47, RPN13-R67, RPN13-Q70, RPN13-R88, UCH1, and UCH2. The sequences of all GST pull-down constructs were verified as correct by DNA sequence analysis.

### 4.7. Yeast Two-Hybrid (Y2H) Analyses

The Y2H vectors used to make the GAL4 AD and BD fusion constructs were pAD-GAL4-2.1 and pBD-GAL4 Cam, respectively (Stratagene, La Jolla, CA, USA). The Y2H analysis was performed as described previously [14]. Yeast culture, transformation, and medium preparation were carried out as described previously [20]. The constructs made for the BD-fused baits are described in Appendix A. These include the following: BD-RPN13, BD-RPN2a, BD-RPN2b, BD-UCH1, BD-UCH2, BD-UCH2-CΔ1, BD-UCH1^C2^, BD-UCH2-A1, BD-UCH2-A2, BD-UCH2-A3, BD-UCH2-A1–2, BD-UCH2-A2–3, BD-UCH2-A1/A3, BD-UCH2-A1–3, BD-UCH1^C2-A2^, and BD-UCH1^C2-A3^. The constructs made for the AD-fused preys are also described in Appendix A and include the following: AD-RPN2a, AD-RPN2b, AD-RPN13, AD-RPN13-22A, AD-RPN13-101A, AD-RPN13-ELAA, AD-RPN13-KDAA, AD-RPN13-ELKD-A4, AD-RPN13-CΔ1, AD-RPN13-CΔ2, AD-RPN13-CΔ3, AD-RPN13-CΔ4, AD-RPN13-CΔ5, AD-RPN13-CΔ6, AD-RPN13-A1, AD-RPN13-A2, AD-RPN13-A3, AD-RPN13-A4, AD-RPN13-A5, AD-RPN13-A4–5, AD-RPN13-A3–5, AD-RPN13-A2–5, AD-RPN13-A1–2, AD-RPN13-A1–3, AD-RPN13-A1–4, AD-RPN13-A1–5, AD-UCH1, and AD-UCH2. The sequences of all the Y2H constructs were verified as correct by DNA sequence analysis. Before conducting pair-wise Y2H analyses, self-activation was first tested for the BD- and AD-fused full-length bait and prey constructs, including BD-RPN2a, BD-RPN2b, BD-RPN13, BD-UCH1, BD-UCH2, AD-RPN2a, AD-RPN2b, AD-RPN13, AD-UCH1, and AD-UCH2. Self-activation was only observed for the BD-fused full-length RPN13 (Appendix A); thus, AD-fused RPN13 or AD-fused RPN13 variants were used for all Y2H analyses involving RPN13 or its variants.

## Figures and Tables

**Figure 1 ijms-25-11650-f001:**
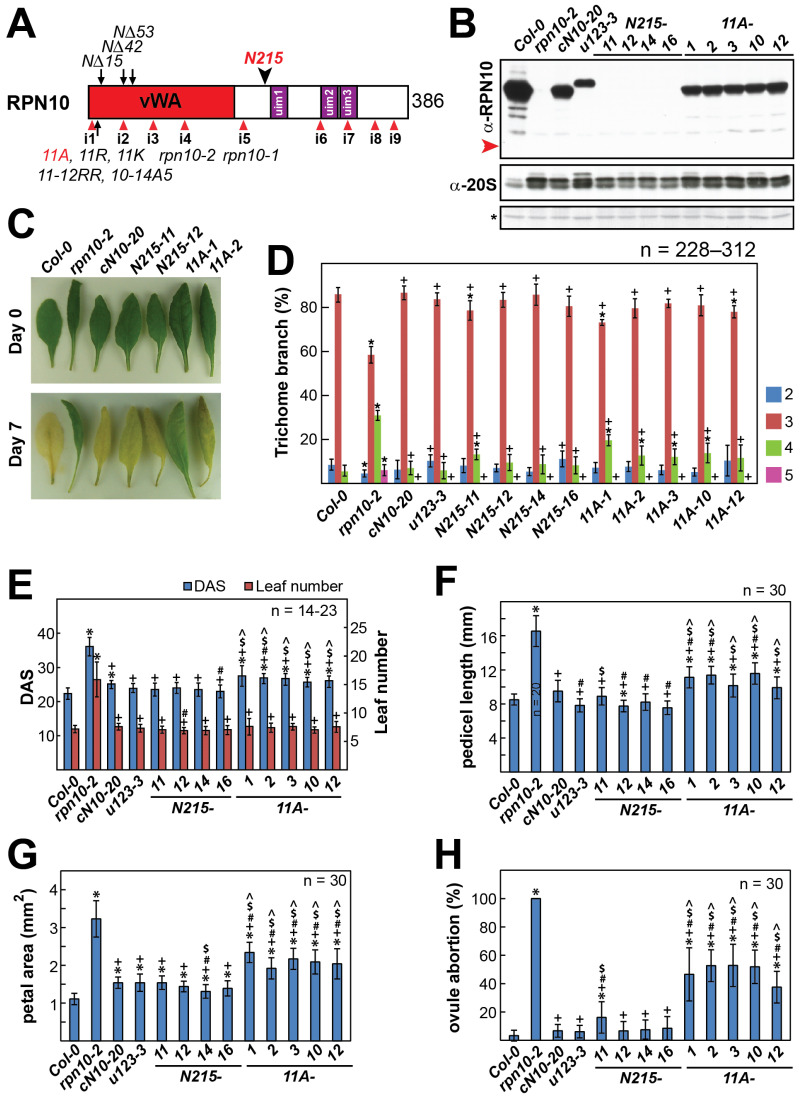
Complementation of growth defects of *rpn10-2* by various deletion and site-specific RPN10 variants. (**A**) A schematic diagram showing truncation points or substitution sites for N215 (black arrow head) and various N-terminal deletion and substitution (black arrows) variants of RPN10, as designated. Corresponding intron sites are indicated by red arrow heads (i1–9). The N-terminal vWA domain and three UIM motifs (UIM1–3) are boxed in solid red and purple boxes. The T-DNA insertion sites for *rpn10-1* and *rpn10-2* are located in intron 5 (i5) and intron 4 (i4), respectively. (**B**) Expression of RPN10 variants in various complemented lines, as indicated. The immunoblot was probed using polyclonal antisera against Arabidopsis RPN10 (α-RPN10). The abundance levels of 20S proteasomes, shown with prominent subunits (the identities of which were not determined), were detected by antisera against 20S proteasomes of *Physcomitrella patens* (α-20S). The * symbol indicates a non-specific band detected by α-20S that served as a loading control. The predicted mobilized position for N215 is indicated by the red arrow head. (**C**) Untreated (Day 0) and dark-induced senescence (Day 7) in 30 DAS rosette leaves from Col-0, *rpn10-2*, and various complemented lines, as indicated. (**D**) The trichome branch number distribution (in percentages) on the upper epidermis of 43 DAS rosette leaves from Col-0, *rpn10-2*, and various *rpn10-2*-complemented lines. Data are derived from four plants and the numbers of trichromes observed with a Stemi SV6 stereomicroscope (Carl Zeiss) are indicated (*n* = 228–312). The classes of different branch numbers (2–5) are color coded as indicated. Significant differences were determined by pairwise comparison with the same number class of trichomes from Col-0 (*, *p* < 0.05) or *rpn10-2* (+, *p* < 0.05) using Student’s *t*-test. (**E**) The flowering times recorded as DAS or rosette leaf numbers for Col-0, *rpn10-2*, and various *rpn10-2*-complementation lines (*n* = 14–23). The flowering time was recorded when the floral stalk reached ~1 cm. Significant differences were determined by pairwise comparison with Col-0 (*, *p* < 0.001), *rpn10-2* (+, *p* < 0.001), cN10 (#, *p* < 0.001), u123 ($, *p* < 0.001), or N215 (^, *p* < 0.0001) using Student’s *t*-test. (**F**) Average pedicel lengths for Col-0 (*n* = 30), *rpn10-2* (*n* = 20), and various *rpn10-2*-complementation lines (*n* = 30). (**G**) Relative petal areas for Col-0, *rpn10-2*, and various *rpn10-2*-complementation lines (*n* = 30, 3 each from 10 flowers). The average area of the Col-0 petals was set to 1. (**H**) The average percentages of aborted ovules from siliques of the Col-0, *rpn10-2*, and various *rpn10-2* complementation lines (*n* = 30, 10 each from 3 plants). (**F**–**H**) Significant differences were determined by pairwise comparison with Col-0 (*, *p* < 0.0001), *rpn10-2* (+, *p* < 0.0001), cN10 (#, *p* < 0.0001), u123 ($, *p* < 0.001), or N215 (^, *p* < 0.0001) using Student’s *t*-test. (**D**–**H**) The error bars represent the SD.

**Figure 2 ijms-25-11650-f002:**
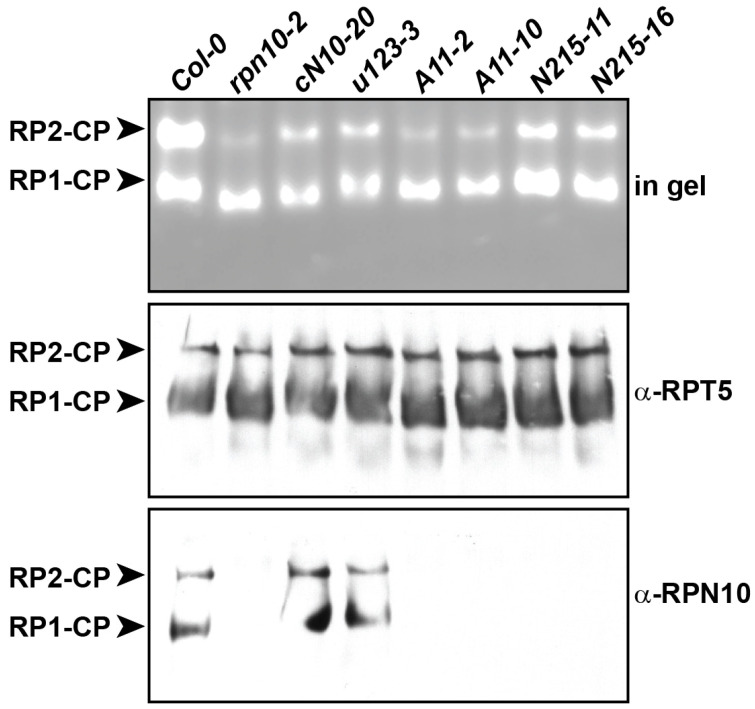
The relative abundances of single- (RP1-CP) and double-capped (RP2-CP) proteasomes and integration of RPN10 variants into the 26S proteasome in complemented *rpn10-2* lines. The relative abundances of single- (RP1-CP) and double-capped (RP2-CP) proteasomes in Col-0, *rpn10-2*, and various *rpn10-2* complementation lines were examined by native PAGE in conjunction with an in-gel activity assay in the presence of 0.02% sodium dodecyl-sulfate (SDS) (in gel) or by immunoblotting using antisera against the base subunit RPT5 (α-RPT5). Integration of RPN10 into single- (RP1-CP) and double-capped (RP2-CP) proteasomes in Col-0, *rpn10-2*, and various *rpn10-2* complementation lines was examined by immunoblotting using antisera against Arabidopsis RPN10 (α-RPN10).

**Figure 3 ijms-25-11650-f003:**
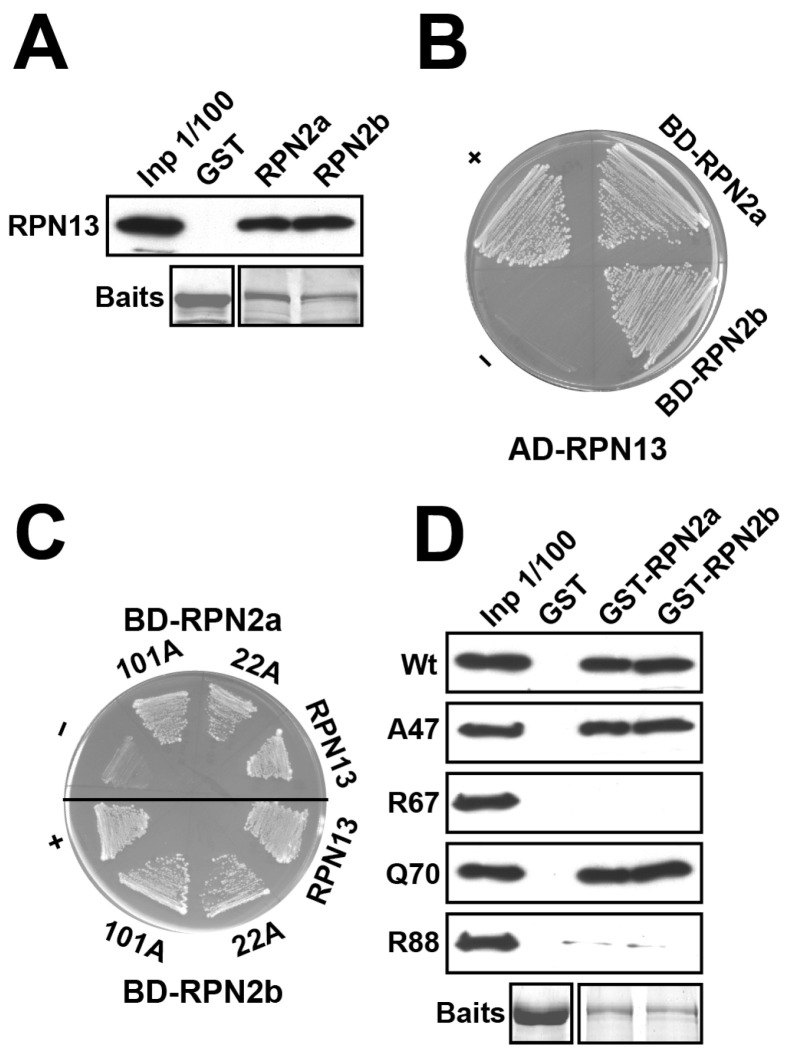
Arabidopsis RPN13 interacts with RPN2a and RPN2b. (**A**) RPN13 is readily pulled down by GST-fused RPN2a or RPN2b. (**B**) Coexpression of AD-fused RPN13 with BD-fused RPN2a or RPN2b activates the *HIS3* reporter, as shown by histidine auxotrophic growth. (**C**) Coexpression of AD-fused wild-type RPN13, site-specific-variant RPN13-101A, or RPN13-22A with BD-fused RPN2a or RPN2b activates the *HIS3* reporter, as shown by histidine auxotrophic growth. (**B**,**C**) Positive (+) and negative (−) controls are p53-SV40 (SV40 T-antigen) and LAMIN (lamin C)-SV40 protein pairs, representing known interacting and non-interacting partners. (**D**) Wild-type RPN13 (Wt) and RPN13 variants A47 and Q70, but not R67 and R88, are readily pulled down by GST-fused RPN2a or RPN2b. (**A**,**D**) Amounts of prey and bait used in pull-down assays are 5 μg for all RPN13 variants and 35 μg, 350 μg, and 320 μg for GST, GST-RPN2a, and GST-RPN2b, respectively. One-hundredth of the input prey (50 ng) and one-tenth of the pull-down products were analyzed by immunoblotting against α-T7. One-tenth of the pull-down products (Baits) was examined by staining with Brilliant Blue R to confirm approximately equivalent immobilization. The pull-down product against GST alone was analyzed as a negative control.

**Figure 4 ijms-25-11650-f004:**
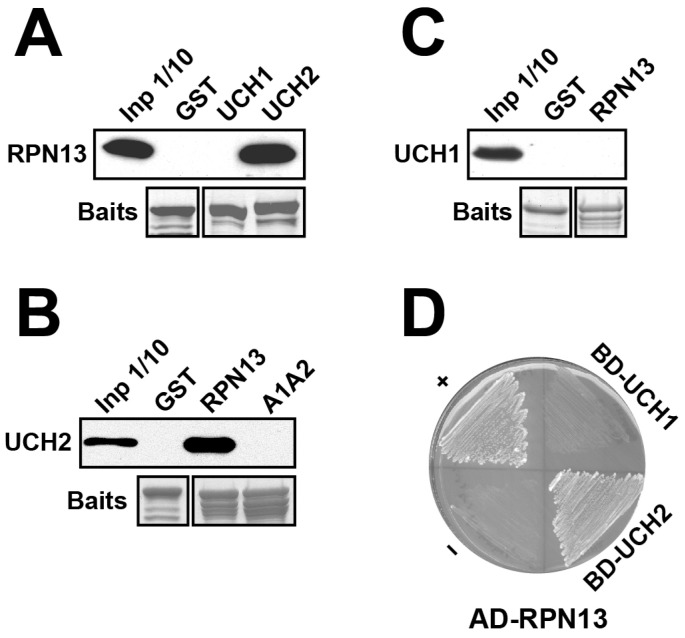
Arabidopsis RPN13 interacts specifically with UCH2 but not with UCH1. (**A**) GST-fused UCH2 but not UCH1 readily pulled down RPN13. (**B**) GST-fused RPN13 but not RPN13-A1A2 readily pulled down UCH2. (**C**) GST-fused RPN13 could not pull down UCH1. (**A**–**C**) Sample amounts used in pull-down assays were 5 μg for all preys (RPN13, UCH1, and UCH2) and ~30 μg, 105 μg, 126 μg, 130 μg, and 130 μg for GST, GST-UCH1, GST-UCH2, GST-RPN13, and GST-RPN13-A1A2, respectively. One-hundredth of the input prey (50 ng) and one-tenth of the pull-down products were analyzed by immunoblotting against α-T7. One-tenth of the pull-down products (Baits) was examined by staining with Brilliant Blue R to confirm approximately equivalent immobilization. The pull-down products against GST alone were analyzed as negative controls. (**D**) Coexpression of AD-fused RPN13 with BD-fused UCH2 but not UCH1 activated the *HIS3* reporter, as shown by histidine auxotrophic growth. The positive (+) and negative (−) controls used were the same as those described in Figure 3.

**Figure 5 ijms-25-11650-f005:**
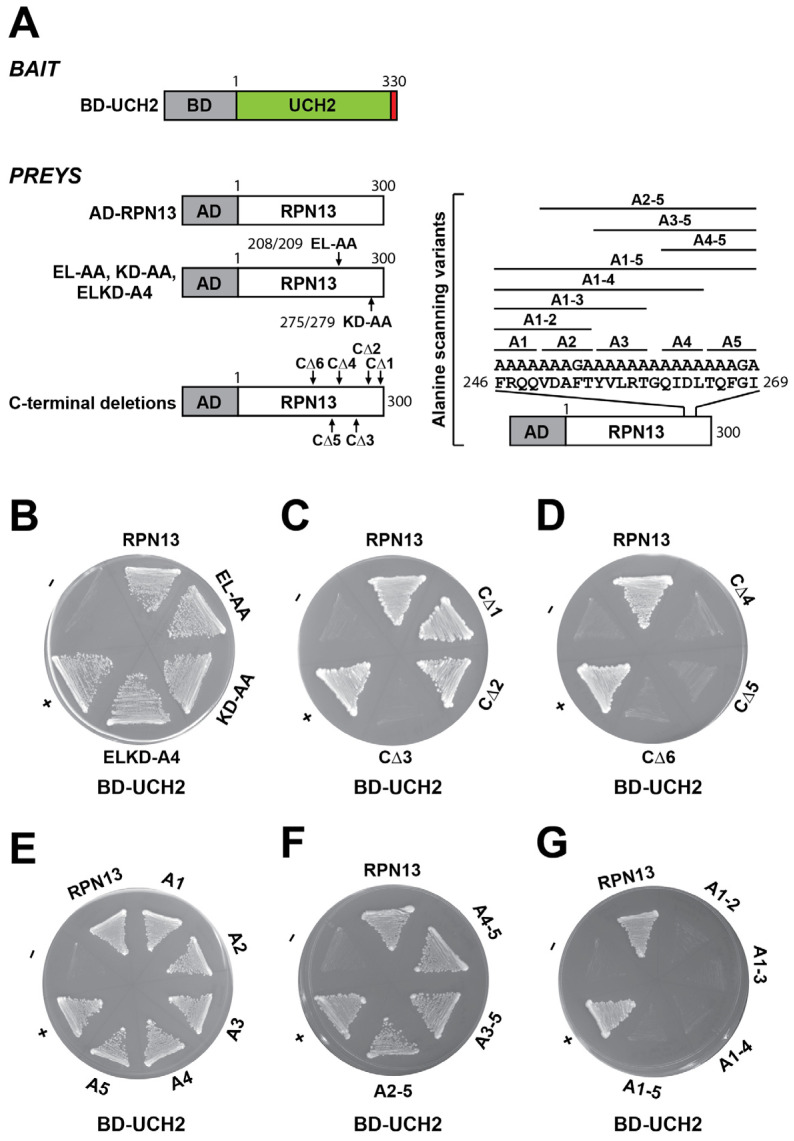
The C-terminal 246–254 region in the Arabidopsis RPN13 DEUBAD domain is critical for UCH2 interactions. (**A**) Schematic diagram shows BD-fused UCH2 (bait) and AD-fused RPN13 variants (preys), including wild-type and site-specific, C-terminal deletion, and alanine scanning mutants for Y2H assays (see the main text for details). The 330-amino-acid UCH2 coding region is boxed in green with its unique 12-amino-acid C-terminal extension designated by a red box. Coordinates for full-length proteins, mutation sites, and alanine scanning region are indicated. (**B**) Four conserved residues, E208, L209, K275, and D279, in Arabidopsis RPN13 are not involved in UCH2 binding. Similar to AD-fused wild-type RPN13, two AD-fused dual-alanine substituted (EL-AA and KD-AA) and one AD-fused quadruple-alanine substituted (ELKD-A4) RPN13 variants could still activate the *HIS3* reporter when coexpressed with BD-fused UCH2. (**C**,**D**) When coexpressed with BD-fused UCH2, only the AD-fused C-terminal deleted RPN13 variants CΔ1 and CΔ2, but not CΔ3–6, could activate the *HIS3* reporter. (**E**) When coexpressed with BD-fused UCH2, similar to AD-fused wild-type RPN13, each of the five subregion-mutated RPN13 variants (A1–A5) could activate the *HIS3* reporter. (**F**) When coexpressed with BD-fused UCH2, similar to AD-fused wild-type RPN13, each of the three serial C-terminally combined subregion-mutated AD-fused RPN13 variants (A4–5, A3–5, and A2–5) could activate the *HIS3* reporter. (**G**) When coexpressed with BD-fused UCH2, unlike AD-fused wild-type RPN13, all four N-terminally combined subregion-mutated AD-fused RPN13 variants (A1–2, A1–3, A1–4, and A1–5) could not activate the *HIS3* reporter. (**B**–**G**) The positive (+) and negative (−) controls used are the same as those described in Figure 3.

**Figure 6 ijms-25-11650-f006:**
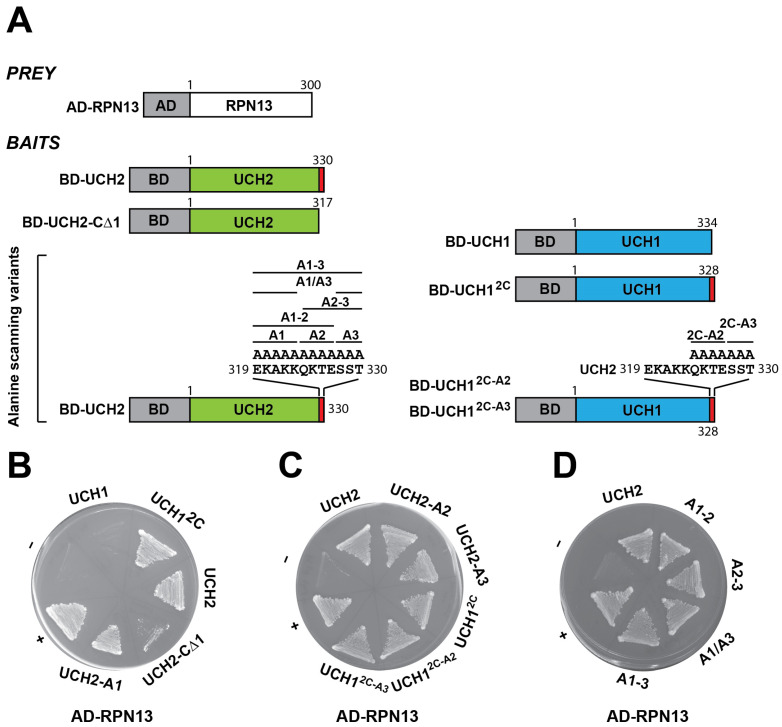
The 12-amino-acid C-terminal extension of Arabidopsis UCH2 is critical for RPN13 interactions. (**A**) Schematic diagram shows AD-fused RPN13 (prey) and BD-fused UCH1 and UCH2 variants (baits), including wild-type (BD-UCH2 and BD-UCH1), a C-terminal-deleted UCH2 (BD-UCH2-CΔ1), alanine scanning UCH2 mutants, and three C-terminally swapped UCH1 mutants (UCH1^2C^, UCH1^2C-A2^, and UCH1^2C-A3^) for Y2H assays (see the main text for details). The UCH2 coding region and its unique 12-amino-acid C-terminal extension are illustrated in the same way as in Figure 5A. The 334-amino-acid UCH1 coding region is boxed in cyan blue. Coordinates for full-length proteins, deletion/swapping sites, and alanine scanning region are indicated. (**B**) When coexpressed with AD-fused RPN13, unlike BD-fused UCH1 (UCH1) and the C-terminal-deleted BD-UCH2 fusion (UCH2-CΔ1), the BD-fused wild-type UCH2 (UCH2), a C-terminal-swapped BD-UCH1 fusion (UCH1^2C^), and a BD-fused UCH2 variant harboring all-alanine substitutions in one of three subregions of the UCH2 C-terminus (UCH2-A1) could activate the *HIS3* reporter. (**C**) When coexpressed with AD-fused RPN13, similar to BD-fused UCH2, two BD-fused UCH2 variants harboring all-alanine substitutions in each of two subregions of the UCH2 C-terminus (UCH2-A2 and UCH2-A3) could activate the *HIS3* reporter. Also, similar to the C-terminal-swapped BD-UCH1 fusion (UCH1^2C^), when coexpressed with the AD-fused RPN13, two C-terminal-swapped BD-UCH1 variants harboring all-alanine substitutions in each of two subregions of the swapped UCH2 C-terminus (UCH1^2C-A2^ and UCH1^2C-A3^) could activate the *HIS3* reporter. (**D**) When coexpressed with AD-fused RPN13, similar to BD-fused UCH2 (UCH2), each of the four BD-fused UCH2 variants harboring all-alanine substitutions in two or all three sub-regions of the UCH2 C-terminus (A1–2, A2–3, A1/A3, and A1–3) could activate the *HIS3* reporter. (**B**–**D**) The positive (+) and negative (−) controls used are the same as those described in Figure 3.

**Table 1 ijms-25-11650-t001:** Altered abundance of 26S proteasome subunits and associated factors in *rpn10-2*
^1^.

Subunit	Locus	*Ratio of Total Number of PSMs* *rpn10-2*/Col-0 (% ± SD) ^2^	Subunit	Locus	*Ratio of Total Number of PSMs* *rpn10-2*/Col-0 (% ± SD)
CP			Base		
PAA1	At5g35590	118.3/100.8 (117.4 ± 8.6) ^3^	RPT1a	At1g53750	141.3/106.6 (132.6 ± 11.8)
PAA2	At2g05840	123.0/89.9 (136.8 ± 6.3)	RPT1b	At1g53780	ND/ND (NA)
PAB1	At1g16470	32.7/43.4 (75.3 ± 80.1)	RPT2a	At4g29040	ND/ND (NA)
PAB2	At1g79210	17.3/ND (NA)	RPT2b	At2g20140	93.3/98.7 (94.5 ± 19.0)
PAC1	At3g22110	102.9/101.3 (101.5 ± 27.4)	RPT3	At5g58290	139.4/118.4 (117.7 ± 16.2)
PAC2	At4g15160	ND/ND (NA)	RPT4a	At5g43010	17.5/16.8 (104.2 ± 180.6)
PAD1	At3g51260	124.0/107.3 (115.6 ± 3.7)	RPT4b	At1g45000	111.8/120.1 (93.1 ± 35.8)
PAD2	At5g66140	32.7/10.5 (312.5 ± 270.9)	RPT5a	At3g05530	145.2/138.2 (105.1 ± 30.4)
PAE1	At1g53850	18.8/11.8 (158.6 ± 141.7)	RPT5b	At1g09100	11.54/ND (NA)
PAE2	At3g14290	26.9/31.6 (85.2 ± 45.1)	RPT6a	At5g19990	44.2/84.2 (52.5 ± 91.0)
PAF1	At5g42790	96.1/107.9 (89.1 ± 13.8)	RPT6b	At5g20000	73.1/ND (NA)
PAF2	At1g47250	10.13/ND (NA)	RPN1a	At2g20580	330.8/272.8 (121.3 ± 20.0)
PAG1	At2g27020	85.6/65.8 (130.1 ± 34.1)	RPN1b	At4g28470	34.6/91.7 (37.7 ± 37.8)
PBA1	At4g31300	90.4/77.6 (116.4 ± 24.2)	RPN2a	At2g32730	157.8/105.0 (150.3 ± 36.6)
PBB1	At3g27430	25.1/39.5 (63.7 ± 65.1)	RPN2b	At1g04810	71.1/44.3 (160.2 ± 66.6)
PBB2	At5g40580	11.9/ND (NA)	RPN10	At4g38630	ND/29.0 (NA) *
PBC1	At1g21720	72.7/89.5 (81.3 ± 11.4)	RPN13	At2g26590	ND/ND (NA)
PBC2	At1g77440	10.9/ND (NA)	RPN15	At1g64750	ND/ND (NA)
PBD1	At3g22630	74.0/31.5 (234.7 ± 64.7)	Lid		
PBD2	At4g14800	6.8/29.7 (23.0 ± 39.8)	RPN3a	At1g20200	157.9/181.6 (86.9 ± 12.5)
PBE1	At1g13060	56.8/34.2 (166.0 ± 31.8)	RPN3b	At1g75990	19.5/48.7 (40.1 ± 40.6)
PBE2	At3g26340	5.21/ND (NA)	RPN5a	At5g09900	51.1/40.5 (126.3 ± 17.5)
PBF1	At3g60820	100.0/100.0 (100.0 ± 25.9)	RPN5b	At5g64760	41.2/21.4 (192.7 ± 172.1)
PBG1	At1g56450	106.7/90.8 (117.6 ± 17.7)	RPN6	At1g29150	158.7/148.7 (106.7 ± 20.4)
26S proteasome-associated factors			RPN7	At4g24820	131.7/117.1 (112.5 ± 23.7)
ECM29	At2g26780	180.8/63.2 (286.2 ± 90.1) *	RPN8a	At5g05780	60.1/89.5 (67.1 ± 19.9)
PA200	At3G13330	89.4/53.9 (165.8 ± 29.8)	RPN8b	At3g11270	39.5/ND (NA)
TPP-II	At4g20850	97.1/152.6 (63.6 ± 20.9)	RPN9a	At5g45620	ND/73.5 (NA) *
			RPN9b	At4g19006	157.7/67.3 (234.4 ± 20.3) *
			RPN11	At5g23540	74.0/59.2 (125.0 ± 63.4)
			RPN12a	At1g64520	97.1/114.5 (84.8 ± 15.4)
			RPN12b	At5g42040	ND/ND (NA)

^1^ Single-capped 20S proteasomes, collected after native gel electrophoresis for PEG-purified 26S proteasomes from Col-0 and *rpn10-2*, were subjected to mass spectrometry analyses. ^2^ The total number of PSMs was based on the amount of subunit-specific peptides detected normalized by the single paralogue subunit PBF1 of the core particle; if common peptides of both paralogues were present, their averaged number of PSMs were partitioned according to the ratio of number of PSMs for subunit-specific peptides before normalization. ND, not detected. ^3^ Numbers in parentheses are percentages of the total number of PSMs of *rpn10-2* in comparison with that of Col-0. NA, not applicable; *, significant according to Student’s *t*-test, *p* < 0.05.

**Table 2 ijms-25-11650-t002:** Segregation rates of homozygous *rpn10-2* in different genotypes.

Genotype	Total Number of Seeds Examined (Germination Rate)	Homozygous *rpn10-2* Number (%)	χ^2^
Col-0 ^1^	605 (99.00%)	NA ^2^	
*N10n10 N13N13*	3610 (99.30%)	69 (1.93)	
*N10n10 n13n13*	1348 (99.50%)	0 (0.00)	
*N10 n10 n13n13* [*N13-1*]	2825 (98.80%)	57 (2.04)	0.17 ^3^
*N10n10 n13n13* [*N13-2*]	3330 (99.73%)	59 (1.78)	0.39
*N10 n10 n13n13* [*R67-1*]	3155 (98.00%)	34 (1.10)	11.49 *
*N10 n10 n13n13* [*R67-2*]	3111 (99.10%)	26 (0.84)	18.82 *
*N10 n10 n13n13* [*Q70-1*]	3124 (99.60%)	77 (2.48)	4.91
*N10 n10 n13n13* [*Q70-2*]	2955 (99.22%)	66 (2.25)	1.45

^1^ Col-0, germination control. ^2^ NA, not applicable. ^3^ Asterisks indicate significant differences detected by χ^2^ using 1.93% as the expected segregation rate of homozygous *rpn10-2* for the *N10n10 N13N13* plants harboring various RPN13 variants (*p* < 0.01).

## Data Availability

Data are contained within the article or the Appendix A.

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
