# Peer review of "The Structural Role of RPN10 in the 26S Proteasome and an RPN2-Binding Residue on RPN13 Are Functionally Important in Arabidopsis"

_ijms, 2024, doi:10.3390/ijms252111650_

Round 1

Reviewer 1 Report

Comments and Suggestions for Authors

General remarks: 

The authors use a combination of vivo functional assays using Arabidopsis, as well as yeast two hybrid (Y2H) and in vitro binding assays to delineate functional domains of proteasomal ubiquitin receptors Rpn13 and Rpn10, as well as of deubiquitinating enzymes UCH1 and UCH2. In line with earlier finding in yeast cells, they find that ubiquitin binding properties of individual Rpn13 and Rpn10 proteins can be mutated without detectable phenotypic consequences, probably due to an extensive redundancy in the detection of ubiquitinated proteins by the proteasome. This study furthermore provides a dissection and delineation of domains of these proteins that are relevant for binding to RPN2 or UCH2 and for proper function in plants. Altogether, this analysis of plant proteasomal ubiquitin receptors contributes to an understanding of their functional domains complementing previous studies.

Additional comments:

Abstract: I was confused by and could not make any sense of the sentence, “Residual segregation of homozygous rpn10-2 was partially compromised for heterozygous rpn10-2 in an rpn13-1 background harboring RPN13-R67, lost RPN2- and ubiquitin-binding, but not RPN13-Q70, lost only ubiquitin-binding, supporting the criticality of RPN13–RPN2 association in vivo.”, when I first read it. Only after reading the details behind this conclusion in the main part of the paper, I understood what it is referring to. To avoid a similar experience by other readers, I would therefore recommend to simplify the conclusion in the abstract, and leave the genetic details of the assay for the main text.

Figure 2: The quality of the scan displaying the in-gel proteasome activity in the top panel is very poor.

Y2H assays:An essential control is missing for the Y2H assays shown in figure 3. For the BD-Rpn2 construct tested, a control where it is combined with an empty AD vector needs to be presented to exclude that the BD construct mediates activation by itself, a problem often encountered with such assays. Without a proper control, these assays do not provide additional value beyond the pulldown experiments. For BD-UCH2, such a problem can be excluded by the experiments shown in figure 5, which show that interaction occurs only with some but not all AD constructs. However, for UCH1-2C and UCH2-As, such controls would also be necessary, as no negative AD controls are included in figure 6 for these constructs.

The paper is extremely long and loaded with many construction details. I would make things easier to follow, if cartoons illustrating the nature of the constructs could be included e.g. in figures 5 and 6. The paragraph extending from lines 238 to 263 can and should be shortened drastically. The discussion should be condensed avoiding extensive repetition of conclusion already provided in the results section.

Comments on the Quality of English Language

Title: "functionally" instead of functional

Author Response

Thank you for your comment, please see the attached file for responses.

Reviewer 2 Report

Comments and Suggestions for Authors

This manuscript provided a comprehensive dissection for the domains and function of RPN10  and RPN13 in the 26S proteasome in Arabidopsis. The authors produced  a variety of truncation or site-specific mutation to test their effects by genetic complementation and analyses of proteasome complex isolation and proteomics by IP/MS. Additionally, they produced a series of mutation in RPN13 and test their efffects on interacting with some proteasome subunits, such as UCH1/2 and RPN2, by Y2H and GST-pull down. The essential residues in RPN10 and RPN13 for the function of 26S proteasome were identified, providing interesting information for future extensive study in 26S proteasome. 

My major concern:
1. The English writing needs to be improved. 

2. The description for the biological materials in main text and figure legends is confusing and unclear. For example, in the Fig. 3D, the two lanes on the right are the same labels of GST-RPN2a. u123, cN10-20,10-14A5,A1A2 are difficult for readers to understand. And what do RPN13-A47/Q70/R67/R88/101A/22A mean? What's original amino acid and what's the mutated amino acid? Typically, the mutaion should be shown like this : K48R, which mean the 48th K is mutated to R.

3. What's the phenotypes in complementation lines of 11R, 11K, 11-12R, and 10-14A5? In Fig. 1C, why the phenotypes of 11A-1 and 11A-2 are opposite, which was inconsistent to the description in maintext?

4. In Fig. 2, the relative ratio of RP2-CP and RP1-CP should be quantified. How many repeats for these assay?

5. ECM29 and PA200 protein levels appeared to be increased in rpn10-2 mutant, is there any other evidence to support it, such as western blot for wild type and rpn10 mutant?

Minor point:

1. In Fig. 3A, the loading of Baits was from the same gel?

Comments on the Quality of English Language

The English is very difficult to understand/incomprehensible. Extensive editing of English is required. 

Author Response

Thank you for your comments, please see the attached file for responses.
